# Stabilizing Zero-Shot Prediction: A Novel Antidote to Forgetting in Continual Vision-Language Tasks

**Zijian Gao**[1,2]†, **Xingxing Zhang**[3]†, **Kele Xu**[1,2]*, **Xinjun Mao**[1,2], **Huaimin Wang**[1,2]

[1]School of Computer, National University of Defense Technology, Changsha, 410000, China.
[2]State Key Laboratory of Complex & Critical Software Environment, Changsha, 410000, China.
[3]School of Computer Science, Tsinghua University, Beijing, 100049, China.
{gaozijian19, xukelele, xjmao, hmwang}@nudt.edu.cn, xxzhang1993@gmail.com

## Abstract

Continual learning (CL) empowers pre-trained vision-language (VL) models to efficiently adapt to a sequence of downstream tasks. However, these models often encounter challenges in retaining previously acquired skills due to parameter shifts and limited access to historical data. In response, recent efforts focus on devising specific frameworks and various replay strategies, striving for a typical learning-forgetting trade-off. Surprisingly, both our empirical research and theoretical analysis demonstrate that the stability of the model in consecutive zero-shot predictions serves as *a reliable indicator* of its anti-forgetting capabilities for previously learned tasks. Motivated by these insights, we develop a novel replay-free CL method named ZAF (Zero-shot Antidote to Forgetting), which preserves acquired knowledge through a zero-shot stability regularization applied to wild data in a plug-and-play manner. To enhance efficiency in adapting to new tasks and seamlessly access historical models, we introduce a parameter-efficient EMA-LoRA neural architecture based on the Exponential Moving Average (EMA). ZAF utilizes new data for low-rank adaptation (LoRA), complemented by a zero-shot antidote on wild data, effectively decoupling learning from forgetting. Our extensive experiments demonstrate ZAF's superior performance and robustness in pre-trained models across various continual VL concept learning tasks, achieving leads of up to 3.70%, 4.82%, and 4.38%, along with at least a 10x acceleration in training speed on three benchmarks, respectively. Additionally, our zero-shot antidote significantly reduces forgetting in existing models by at least 6.37%. Our code is available at `https://github.com/Zi-Jian-Gao/Stabilizing-Zero-Shot-Prediction-ZAF`.

## 1 Introduction

In the rapidly evolving landscape of artificial intelligence, pre-trained models have become fundamental to achieving state-of-the-art results across a myriad of applications [25]. Recently, large vision-language (VL) models have provided remarkable predictions on downstream tasks without any training examples [30, 13, 2, 19]. However, these models are typically trained on static datasets, which may not capture the continuously evolving variety and complexity of real-world data. As new concepts emerge and existing categories expand, the static nature of these pre-trained models can lead to diminished performance over time. For example, the CLIP model [30] achieves an accuracy of less than 60% on the MNIST dataset (i.e., a performance significantly lower than that of a conventionally trained CNN [17]) [49]. To bridge this gap, continual learning (CL) has emerged as a vital methodology, making learning new knowledge a lifelong process for the pre-trained VL model.

---

*Corresponding author. † Equal Contribution.

38th Conference on Neural Information Processing Systems (NeurIPS 2024).

However, the adaptation process, whether involving comprehensive fine-tuning of the entire pre-trained model [7, 49, 3] or parameter-efficient continual fine-tuning [33, 45, 29], inevitably undergo incremental parameter shifts. This presents a significant challenge in CL, where VL models forget historical knowledge as they adapt to new tasks [41, 45]. Additionally, the high computational costs required to integrate new data with old data, combined with often limited access to complete previous datasets, further exacerbate the severity of forgetting [47, 45]. Thus far, most CL research has focused on mitigating the forgetting of class information from previously learned images, a process known as class incremental learning [38]. Multimodal tasks such as visual question answering (VQA) and natural language visual reasoning (NLVR) have received less attention. Building upon the foundations laid by unimodal CL efforts, recent initiatives have aimed to mitigate forgetting from parameter shifts by developing robust continual VL learning frameworks, including Mixture-of-Experts [45], weight consolidation [49], and Layered-LoRA [33]. Furthermore, a growing body of work is dedicated to developing anti-forgetting strategies, such as knowledge distillation [23] and rectification [3]. Meanwhile, latent replay mechanisms, including adversarial pseudo-replay [33] and prototype learning [47], have shown effectiveness in retaining historical knowledge, albeit with significant additional memory and computational costs. All these methods, however, struggle with the inherent trade-off between acquiring new knowledge and preserving historical knowledge.

In this work, we aim to decouple the learning-forgetting dynamic, striving for *a win-win outcome* — enhancing anti-forgetting capabilities without interfering with the learning process. Our extensive empirical investigation reveals *uniform patterns* in the model's performance across learned and future VL tasks. Theoretically, we find that the generalization errors for old and new tasks are nearly identical. Motivated by these insights, particularly the observed stability of zero-shot predictions that reflects the stability required for old tasks (i.e., anti-forgetting capabilities), we develop a novel CL approach. This method, named **ZAF** (Zero-shot Antidote to Forgetting), incorporates a zero-shot prediction stability regularization term within our EMA-LoRA architecture. ZAF employs two low-rank adapters: one facilitates the learning of new tasks and ensures stable zero-shot predictions on wild data; the other, an Exponential Moving Average (EMA)-based adapter, preserves historical knowledge and utilizes it for zero-shot supervision. The synergistic interaction between these adapters effectively decouples learning from forgetting, thereby enhancing overall CL performance.

Our contributions include: (1) We conduct a comprehensive empirical and theoretical study on continual learning for pre-trained VL models, establishing zero-shot stability as a reliable indicator of anti-forgetting capabilities; (2) Inspired by this finding, we develop a plug-and-play zero-shot antidote that enhances models' anti-forgetting across various CL methods, through a zero-shot prediction stability regularization on wild data; (3) We introduce an innovative replay-free CL method (termed ZAF), implemented within a parameter-efficient EMA-LoRA architecture, effectively decoupling learning from forgetting; (4) Across various continual VL benchmarks and pre-trained models, our approach achieves state-of-the-art performance and significantly reduced complexity.

## 2 Related Work

**Continual Learning (CL):** Continual learning aims to train a single model capable of incrementally updating its knowledge with a new sequence of tasks while preserving historical knowledge [41, 39]. However, due to parameter shift and limited access to historical data, the primary challenge in CL is forgetting previously learned tasks over time [41]. A plethora of strategies have been explored to address this issue [32, 21, 18, 6, 12], including selective stabilization of network weights, replay of a few old training samples, construction of task-specific parameters, etc. Despite advancements in the unimodal community, such as in image classification, multimodal tasks have received comparatively less attention. This gap highlights the complexity and emerging interest in multimodal settings [9, 4, 8, 24, 35, 47]. In this context, VQACL [47] introduced a continual VQA task leveraging a latent replay strategy. RATT [4] focused on continual image captioning through weight regularization and knowledge distillation. Distinctively, ConStruct-VL [33] pursued continual adaptation to fine-grained Structured VL Concepts reasoning skills without fixed task boundaries. This is particularly applicable to real-world scenarios, and thus, we choose this VL task as *a case study* in our experiments.

It is noteworthy that several studies closely related to ours discuss both zero-shot and continual learning capabilities [49, 45, 46]. However, their objectives differ fundamentally from ours. These studies, including ZSCL [49] and MoE-Adapters [45], aim to preserve the zero-shot transfer ability inherent in pre-trained VL models during continual adaptation for sequentially arriving tasks, even

though these capabilities may be inherently limited. In contrast, our research is exclusively focused on enhancing the anti-forgetting capabilities in downstream CL tasks with our zero-shot antidote.

**Vision-and-Language (VL) Pre-training:** VL pre-training aims to enhance the performance of downstream VL tasks by pre-training models on large-scale image-text data. The effectiveness of these models largely depends on their representational capacity, including factors like data quality and model architecture, as well as the similarity between pre-training tasks and downstream applications. Generally, higher data quality and greater task similarity lead to stronger generalizability. However, due to the high costs of human annotation, most methods leverage noisy image-text pairs sourced from the web, which are a sub-optimal form of supervision (e.g., CLIP [30] and ALIGN [13]). A novel dataset bootstrapping method, Captioning and Filtering (CapFilt), was introduced to utilize web datasets more effectively [19]. Furthermore, most existing pre-trained models excel either in understanding-based tasks like image-text retrieval [30, 20] or generation-based tasks such as image captioning [26, 11]. The primary challenge lies in designing model architectures capable of performing diverse tasks [44]. In response, the 'BLIP' framework was proposed, providing flexible transfer ability to both tasks [19]. Specifically, its image transformer is initialized from ViT pre-trained on ImageNet [37, 5], and the text transformer originates from BERT [14]. An important variant of BLIP, enhanced with CapFilt to boost performance, is termed 'BLIP w/ CapFilt-L' [19]. Additionally, to enable reasoning on downstream tasks, 'BLIP' was further fine-tuned on the NLVR2 dataset [36] with a more computationally efficient architecture, referred to as 'BLIP w/ NLVR' [19].

In this work, we primarily focus on BLIP and its two variants, unlike the closely related studies [49, 45, 46] that utilize CLIP. Our choice is driven by the nature of our case study, which involves reasoning tasks as opposed to classification tasks typically associated with CLIP.

## 3 Preliminary Analysis

In this section, we first define the problem of continual VL learning and then examine the correlation between the model's anti-forgetting capabilities and zero-shot stability through an empirical study.

### 3.1 Formulation of Continual Vision-Language (VL) Learning

In this work, we focus on adapting pre-trained VL models to a sequence of newly arriving reasoning tasks, denoted as $\{\mathcal{T}^1, \cdots, \mathcal{T}^n\}$. Each task $\mathcal{T}^t$ involves a set of image-text pairs $(I, T)$, with a binary ground-truth function $\overline{P}(T, I) \in \{(0, 1), (1, 0)\}$ determining whether the text precisely describes the image content. The primary distinction among these tasks lies in the types of cognitive skills they aim to develop. Notably, traditional CL methods, which require identifying the specific task to which input belongs during inference, are impractical here. This is due to the challenge of discerning the required skills from free-form text and the likelihood that multiple skills may be simultaneously necessary. Given the constraints of privacy in practical applications, we adopt a strict policy of not retaining any task-specific data between training sessions. This leads us to a *continual*, *replay-free* setting without task identifiers, which is precisely the focus of our study.

### 3.2 Empirical Study of Anti-Forgetting and Zero-Shot Performance

To investigate the correlation between a model's anti-forgetting capabilities and its stability in zero-shot predictions, we conducted an empirical study using the Structured VL Concepts (SVLC) learning multimodal benchmark [33]. This benchmark utilizes two widely recognized public VL datasets: Visual Genome (VG) [16] and Visual Attributes in the Wild (VAW) [28], adhering to protocols outlined by the VL-Checklist [48]. The '**7 Task VG+VAW**' benchmark, derived from both VG and VAW datasets, includes seven distinct concepts designed to assess a VL model's understanding of various relationships, including spatial and inter-object transitive actions, as well as attributes such as size, color, material, and intransitive single-object actions. Examples of these concepts are depicted in Fig. 5. Additionally, the '**7 Task VG**' benchmark is solely based on the VG dataset, and the '**5 Task VAW**' benchmark focuses on five attributes and object state concepts from the VAW dataset. We adhere strictly to the dataset splits outlined in ConStruct-VL [33].

We rigorously evaluate the official implementations of all baseline models to ensure a fair comparison. Our evaluations largely adhere to the training regimes established by LoRA [10], MoE-Adapters [45],

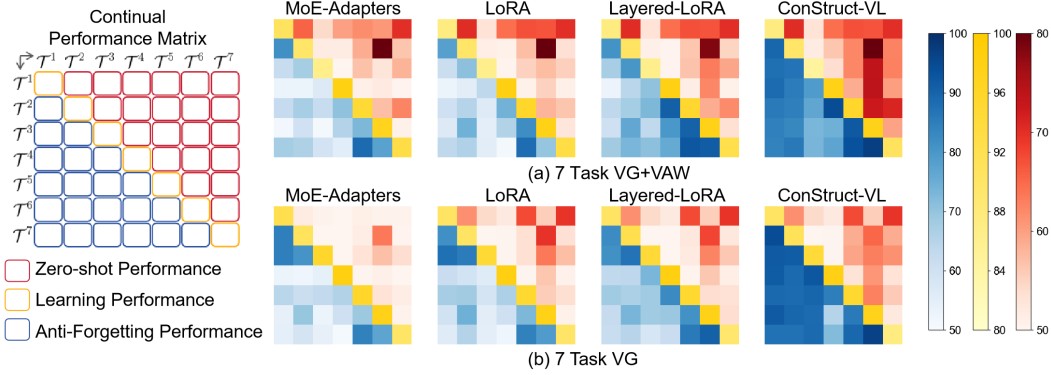

Figure 1: Empirical study of anti-forgetting, learning, and zero-shot performance in CL methods.

Layered-LoRA [33], and ConStruct-VL [33]. Since MoE-Adapters [45] was originally designed for both pre-trained and downstream classification tasks requiring task-id inference, we adapt its implementation to include a single task-dependent router for our needs. Given the focus on reasoning tasks, we opt for 'BLIP w/ NLVR' as our pre-trained model, chosen for its robust representation capabilities and adequate zero-shot transfer ability.

Fig. 1 illustrates a heatmap that depicts the performance of various CL methods across two benchmarks (see Appendix C.1 for additional benchmarks). In this heatmap, rows represent training steps, and $A_{ij}$ denotes the prediction accuracy on task $\mathcal{T}^j$ after training on task $\mathcal{T}^i$. It is important to note that traditional CL methods typically focus only on the *lower triangular matrix* of results, which represents performance on previously learned tasks. However, our analysis extends to include zero-shot predictions on future tasks, which are represented in the *upper triangular matrix*. By comparing these results, we observe that larger average values and less fluctuation in the red area, which represents strong zero-shot stability, typically correspond to similar patterns in the blue area, indicative of strong anti-forgetting capabilities, without adversely affecting the yellow area, where learning new tasks occurs. This observation suggests that a model's stability in zero-shot predictions can reflect its anti-forgetting capabilities. Our findings further suggest that by systematically stabilizing zero-shot predictions during continual learning, we can significantly enhance the model's ability to retain historical knowledge without compromising the acquisition of new information.

# 4 Theoretical Foundation and Our Approach

In this section, we first provide a theoretical analysis to substantiate our empirical findings. Subsequently, we present an innovative CL approach specifically tailored for pre-trained VL models.

## 4.1 Continual Vision-Language Learning Objective

In the continual learning of sequentially arriving VL tasks $\mathcal{T}^t = \{T, I\}$, the model is designed to align with the binary ground-truth function $\overline{P}(T, I)$ that evaluates whether the text $T$ precisely describes the image content $I$. In this case study, we employ BLIP-based encoders to learn discriminative deep embeddings. Specifically, let $f_\nu$ represent the image encoder and $f_\tau$ the text encoder, while $h_\omega$ and $h_\psi$ serve as the cross-attention decoder and the final binary classifier, respectively. The predictive modeling process within the BLIP framework for learning task $\mathcal{T}^t$ is structured as follows:

$$P^t(T, I) = h_{\psi^t}(h_{\omega^t}(f_{\nu^t}(I), f_{\tau^t}(T))), \tag{1}$$

where $P^t(T, I)$ denotes the prediction probability made by the current model $\mathcal{M}^t = \{\nu^t, \tau^t, \omega^t, \psi^t\}$.

To sequentially fine-tune the BLIP model for downstream tasks, we employ a cross-entropy loss term, $\mathcal{L}_{CE}(P^t(T, I), \overline{P}(T, I))$. This loss measures the discrepancy between the current prediction, $P^t(\cdot)$, and the ground truth, $\overline{P}(\cdot)$, ensuring that the model progressively acquires new information.

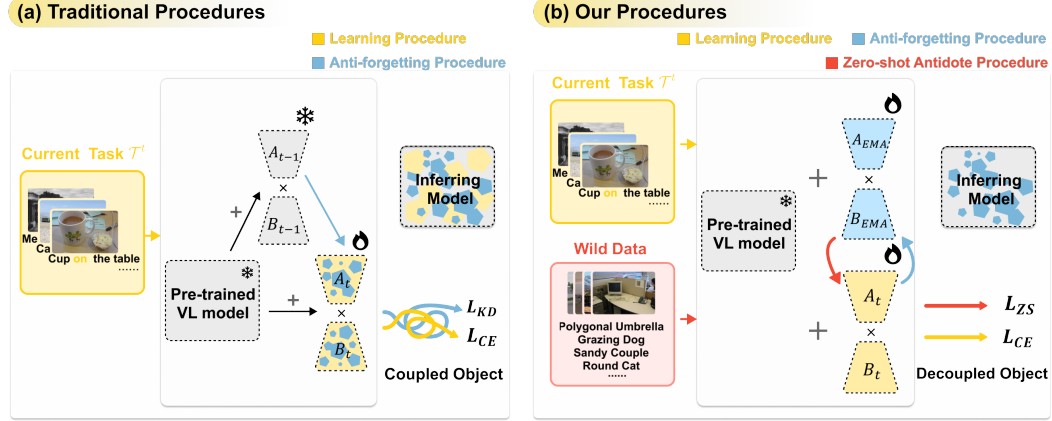

Figure 2: Comparison of training and inference procedures between traditional and our CL methods.

Let $\hat{\mathcal{E}}_{1:t}$ denote the empirical errors on the $t$ observed tasks, and $\hat{\mathcal{E}}_t$ represent the empirical error on the current task $\mathcal{T}^t$. For $k \in \{t+1, \ldots, n\}$ and $s \in \{1, \ldots, t-1\}$, $\mathcal{E}_k$ and $\mathcal{E}_s$ denote the generalization errors on the future task $\mathcal{T}^k$ and the old task $\mathcal{T}^s$, respectively. Inspired by the PAC-Bayes theory [27] and previous work in domain generalization [1, 31, 40], we present the upper bounds of these two errors under the continual VL scenario (see Appendix A for the detailed proof).

**Proposition 1** *For continual learning with pre-trained VL models, let $\mathcal{M}^t$ denote a solution of the continually learned tasks $\mathcal{T}^1, \cdots, \mathcal{T}^t$. In particular, $\mathcal{M}^t = \arg\min_{||\mathcal{M}-\mathcal{M}^{t-1}||_2 \leq \Delta} \hat{\mathcal{E}}_t(\mathcal{M})$ where $||\mathcal{M} - \mathcal{M}^{t-1}||_2 \leq \Delta$ represents the weight vectors for continual tasks are only minor variations. For any $\delta \in (0, 1)$ with probability at least $1 - \delta$:*

$$\forall s \in \{1, \cdots, t-1\}, \quad \mathcal{E}_s(\mathcal{M}^t) \leq \hat{\mathcal{E}}_{1:t}(\mathcal{M}^t) + \frac{1}{2t}\sum_{i=1}^t \mathrm{Div}(\mathcal{T}_i, \mathcal{T}_s) + \sqrt{\frac{d[\ln(\bar{N}/d)] + \ln(1/\delta)}{2\bar{N}}}, \quad (2)$$

$$\forall k \in \{t+1, \cdots, n\}, \quad \mathcal{E}_k(\mathcal{M}^t) \leq \hat{\mathcal{E}}_{1:t}(\mathcal{M}^t) + \frac{1}{2t}\sum_{i=1}^t \mathrm{Div}(\mathcal{T}_i, \mathcal{T}_k) + \sqrt{\frac{d[\ln(\bar{N}/d)] + \ln(1/\delta)}{2\bar{N}}}, \quad (3)$$

*where $\mathrm{Div}(\mathcal{T}_i, \mathcal{T}_j) := 2\sup_{h \in \mathcal{H}} |\mathcal{P}_{\mathcal{T}_i}(I(h)) - \mathcal{P}_{\mathcal{T}_j}(I(h))|$ defines the $\mathcal{H}$-divergence between the distributions for tasks $\mathcal{T}_i$ and $\mathcal{T}_j$, with $I(h)$ being the characteristic function. Here, $\bar{N}$ denotes the harmonic mean of the training example sizes for the $t$ observed tasks, and $d$ represents the VC dimension of the parameter space.*

Proposition 1 demonstrates that the model $\mathcal{M}^t$ has consistent upper bounds on the generalization errors for both previously learned and future tasks. These abilities are significantly influenced by three key factors: *empirical error of continual tasks*, *discrepancy between task distributions*, and *complexity of the parameter space*. Such consistency underscores that the model's capabilities in **zero-shot prediction** can reliably indicate its **anti-forgetting** capabilities. In practice, maintaining this consistency necessitates the implementation of carefully designed regularization techniques.

Motivated by these insights, we develop a zero-shot regularization antidote to mitigate forgetting. Considering the unpredictable nature and broad array of future tasks, we initially introduce an unlabeled wild dataset, denoted as $\mathcal{D}_{\mathrm{wild}} = \{T_{\mathrm{wild}}, I_{\mathrm{wild}}\}$, to assess the zero-shot capabilities of the continually learned VL model. It is crucial to note that both the text $T_{\mathrm{wild}}$ and the images $I_{\mathrm{wild}}$ are entirely unpaired and unlabeled[2]. Additionally, the substantial number of wild examples in $\mathcal{D}_{\mathrm{wild}}$ are distinct and separate from the actual downstream tasks $\{\mathcal{T}^i\}_{i=1}^n$. Fig. 8 illustrates some wild examples.

To stabilize zero-shot predictions, we develop a novel loss term, $\mathcal{L}_{\mathrm{ZS}}(P^t(T_{\mathrm{wild}}, I_{\mathrm{wild}}), \widehat{P^t}(T_{\mathrm{wild}}, I_{\mathrm{wild}}))$, where $\widehat{P^t}(\cdot)$ is the prediction probability derived from the model in the previous training step and used for zero-shot supervision. The formulation of our method's loss function is outlined below:

$$\mathcal{L} = \mathcal{L}_{\mathrm{CE}}(P^t(T, I), \bar{P}(T, I)) + \mathcal{L}_{\mathrm{ZS}}(P^t(T_{\mathrm{wild}}, I_{\mathrm{wild}}), \widehat{P^t}(T_{\mathrm{wild}}, I_{\mathrm{wild}})), \quad (4)$$

---

[2]Please refer to Appendix C.5 for more details about the construct of the wild dataset.

---
**Algorithm 1** Training Algorithm of ZAF
---
**Input**: A pre-trained VL model $\mathcal{M}^0$, number of epochs $E$, and hyperparameter $\alpha \in [0, 1]$
**Data**: A sequence of training tasks $\mathcal{T}^1, ..., \mathcal{T}^n$ and a wild dataset $\mathcal{D}_{\text{wild}} = \{T_{\text{wild}}, I_{\text{wild}}\}$
**Output**: The continually learned model $\mathcal{M}^t$ for inference

  1: Initialize the LoRA component $\{\mathcal{A}, \mathcal{B}\}$ and EMA component $\widehat{\mathcal{W}}$
  2: **for** $t$ in $1 : n$ **do**
  3:     **for** $e$ in $E$ **do**
  4:         **while** not traverse over all current data $\mathcal{T}^t$ **do**
  5:             Sample a batch of current data $\mathcal{B}^i$, and a batch of wild data $\mathcal{B}^{\text{wild}}$
  6:             $\mathcal{W} = \mathcal{M}^0 + \mathcal{A} \cdot \mathcal{B}$
  7:             Predict the probabilities $P^t(\mathcal{B}^i)$ using $\mathcal{W}$
  8:             Calculate $\mathcal{L}_{\text{CE}}$ with $P^t(\mathcal{B}^i)$ and ground truth $\overline{P}(\mathcal{B}^i)$
  9:             **if** $t = 1$ **then**
10:                 Optimize $\{\mathcal{A}, \mathcal{B}\}$ with $\mathcal{L}_{\text{CE}}$
11:             **else**
12:                 Predict the probabilities $P^t(\mathcal{B}^{\text{wild}})$ using $\mathcal{W}$, and $\widehat{P}^t(\mathcal{B}^{\text{wild}})$ using $\mathcal{M}^{t-1}$
13:                 Calculate $\mathcal{L}_{\text{ZS}}$ with $P^t(\mathcal{B}^{\text{wild}})$ and $\widehat{P}^t(\mathcal{B}^{\text{wild}})$
14:                 Optimize $\{\mathcal{A}, \mathcal{B}\}$ with loss function Eq. (4)
15:         **if** $t = 1$ **then**
16:             $\widehat{\mathcal{W}} \leftarrow \mathcal{A} \cdot \mathcal{B}$, and $\mathcal{M}^1 = \mathcal{M}^0 + \widehat{\mathcal{W}}$
17:         **else**
18:             Update the EMA component $\widehat{\mathcal{W}}$ and $\mathcal{M}^t$ with Eq. (5)
19: **return** $\mathcal{M}^t$
---

where $\mathcal{L}_{\text{ZS}} = ||P^t(T_{\text{wild}}, I_{\text{wild}}) - \widehat{P^t}(T_{\text{wild}}, I_{\text{wild}})||_1$ in our experiments to restrain its fluctuations.

## 4.2 EMA-LoRA Architecture

To achieve the objective outlined above, rather than conducting comprehensive fine-tuning of the entire pre-trained model $\{\nu^t, \tau^t, \omega^t, \psi^t\}$ as defined in Eq. (1), we develop a parameter-efficient approach using an EMA-LoRA architecture based on BLIP. As depicted in Fig. 2(b), for efficient adaptation to new tasks, we first integrate LoRA adapters into all layers of the image encoder $f_\nu$, text encoders $f_\tau$, and the cross-attention decode $h_\omega$, which are updated during the training of task $\mathcal{T}^t$. Specifically, each of $f_\nu$, $f_\tau$, and $h_\omega$ consists of a combination of non-parametric functions - referred to here as data norms, such as LayerNorm, which remain frozen, and two types of parametric functions, namely linear and embedding functions. Typically, linear and embedding functions are parameterized by a weight matrix $\mathcal{W}$, which is optimized at every iteration by adding residuals to the weights from the initial pre-trained VL model $\mathcal{M}^0$. Formally, this is represented as $\mathcal{W} = \mathcal{M}^0 + \mathcal{A} \cdot \mathcal{B}$, where $\mathcal{A}$ and $\mathcal{B}$ are learnable low-rank matrices of dimensions $m \times r$ and $r \times l$ respectively, with $m \times l$ being the dimensions of $\mathcal{M}^0$. During the training of task $\mathcal{T}^t$, only the current task's LoRA adapters $\{\mathcal{A}, \mathcal{B}\}$ are learned, and predictions $P^t(\cdot)$ are made using $\mathcal{W}$. This strategy not only enhances the adaptability of the model to new tasks but also maintains a balance between efficiency and performance.

As demonstrated in Eq. (4), accessing previous models is essential for predicting wild data (i.e., $\widehat{P}(\cdot)$). To facilitate this, we implement the Exponential Moving Average (EMA) on the aforementioned LoRA adapters (see Fig.2(b)). This approach allows for memory-efficient access to previous models while learning new tasks. Specifically, the EMA process is conducted at the end of every epoch:

$$\widehat{\mathcal{W}} \leftarrow \alpha\widehat{\mathcal{W}} + (1 - \alpha)\mathcal{A} \cdot \mathcal{B}, \quad \text{and} \quad \mathcal{M}^t = \mathcal{M}^0 + \widehat{\mathcal{W}}, \tag{5}$$

where the hyperparameter $\alpha \in [0, 1]$ plays a crucial role in updating the model parameters (refer to Fig. 3 for its sensitivity analysis). Importantly, $\mathcal{M}^t$ represents the final model used for inference after continual learning. The weights $\widehat{\mathcal{W}}$ are updated following the completion of the first downstream task, at which point there is no risk of forgetting, thereby eliminating the need for the zero-shot antidote $\mathcal{L}_{\text{ZS}}$. For subsequent tasks, predictions on wild data using $\widehat{P^t}(\cdot)$ are made with $\mathcal{M}^{t-1}$.

Algorithm 1 provides a detailed view of our training algorithm for the Zero-shot Antidote to Forgetting (ZAF) framework. Key highlights of the proposed ZAF include: (1) Learning vs. Forgetting: As

illustrated in Fig. 2(a) and (b), traditional CL frameworks, such as those incorporating knowledge distillation, often struggle with a trade-off between learning and forgetting. This trade-off typically manifests in the balance of $\mathcal{L}_{\text{CE}}$ and $\mathcal{L}_{\text{KD}}$, where the LoRA adapter is optimized to align with both old and new knowledge using current data: $\min \mathcal{L}_{\text{CE}}(P^t(\mathcal{T}^t), \overline{P}(\mathcal{T}^t)) + \mathcal{L}_{\text{KD}}(P^t(\mathcal{T}^t), P^{t-1}(\mathcal{T}^t))$. Our approach introduces a zero-shot antidote using wild data to effectively separate the learning processes from the problem of forgetting: $\min \mathcal{L}_{\text{CE}}(P^t(\mathcal{T}^t), \overline{P}(\mathcal{T}^t)) + \mathcal{L}_{\text{ZS}}(P^t(\mathcal{D}_{\text{wild}}), \widehat{P}^t(\mathcal{D}_{\text{wild}}))$. (2) Privacy Protections: Unlike conventional methods that require replaying old task data, our framework utilizes generated wild data to prevent forgetting, adhering to privacy constraints. (3) Innovative Mixture Strategy: Deviating from the typical *spatial* Mixture of Experts (MoE) and Layered-LoRA architectures, our model employs a *temporal* mixture strategy through EMA. This approach not only conserves memory but also integrates downstream tasks incrementally and effectively. (4) Efficiency in Resource Usage: The EMA-LoRA architecture involves only two low-rank adapters, which are computationally efficient, and also ensure $||\mathcal{M} - \mathcal{M}^{t-1}||_2 \leq \Delta$, as established in our Proposition 1.

## 5 Experiment

**Benchmark:** We utilize the '7 Task VG+VAW', '7 Task VG', and '5 Task VAW' benchmarks for our empirical studies as discussed in Sec. 3.2. We use the $object\ state \rightarrow attr.\ action \rightarrow attr.\ size \rightarrow rel.\ spatial \rightarrow attr.\ material \rightarrow rel.\ action \rightarrow attr.\ color$ 7-task sequence in the 7 Task VG+VAW benchmark and 7 Task VG benchmark, and the $object\ state \rightarrow attr.\ action \rightarrow attr.\ size \rightarrow attr.\ material \rightarrow attr.\ color$ 5-task sequence in the 5 Task VAW benchmark. Given the reasoning task scenario, our primary focus is on the original 'BLIP' model and its two variants: 'BLIP w/ CapFilt-L' and 'BLIP w/ NLVR'. Notably, the latter variant, which is fine-tuned on the NLVR2 dataset, exhibits enhanced zero-shot transfer ability.

**Model Architecture:** All experiments begin with the BLIP model architectures and its pre-trained weights. The image encoder, denoted as $f_\nu$, utilizes ViTB/16, and the text encoder $f_\tau$ is a BERT with a 12-layer encoder and 768 hidden size. The decoder $h_\omega$ extends $f_\tau$, incorporating cross-attention layers between every two self-attention layers, with each receiving encoded image tokens as additional input. Our binary classifier $h_\psi$ is a 2-layer MLP with a hidden size of 768. For all baselines, we adhere to the implementation protocols established in ConStruct-VL [33], a pioneering work in SVLC scenario that has demonstrated exceptional performance. We maintain a shared binary classifier $h_\psi$ across all task models. In practice, for the BLIP w/ NLVR model, $h_\psi$ is frozen across all tasks. For the BLIP and BLIP w/ CapFilt-L models, $h_\psi$ is trained only for the first task $\mathcal{T}^1$ and then remains frozen for all subsequent tasks.

**Baseline and Implementation:** Due to the inadequate performance of unimodal CL methods in multimodal scenarios, as highlighted by [33], our analysis primarily focuses on frameworks and multi-modal CL methods designed for VL scenarios. We include only one representative unimodal method, LwF [21], which mitigates forgetting through knowledge distillation for comparison. Detailed comparisons with other unimodal CL methods are provided in Appendix C.6. Among the methods compared, Continual-FT [7] and LoRA [10] train the VL model continuously without incorporating anti-forgetting procedures. Layered-LoRA [33] and MoE-Adapters [45] are viewed as representative continual VL learning frameworks. Furthermore, we compare our ZAF with ZSCL [49], which aims to preserve the inherent zero-shot capability of the pre-trained model, and ConStruct-VL [33], which employs an expensive pseudo-replay strategy involving adversarial attacks on every sample. Following the implementation from previous work [33], specifically, we adapt WD 0.05, an initial LR of 1.25e-3, a cosine scheduler, and a maximum of 12 training epochs in all experiments. For all low-rank adapters, the rank $r$ is set to 16. The only hyperparameter, $\alpha$, of our method is fixed at 0.85. Please refer to Appendix B for more implementation details.

**Evaluation Metrics:** We utilize three widely-recognized evaluation metrics for continual learning - Final Average Accuracy (FAA), Cumulative Average Accuracy (CAA), and Final Forgetting Measure (FFM) as detailed in [41, 38]. We define the accuracy on the task $\mathcal{T}^j$ after learning the task $\mathcal{T}^i$ as $A_{ij}$. The average accuracy after learning task $\mathcal{T}^i$ is denoted as $AA_i = \frac{1}{i}\sum_{j=1}^{i} A_{ij}$. Upon completing all $n$ tasks, we report FAA $= AA_n$, CAA $= \frac{1}{n}\sum_{i=1}^{n} AA_i$, and FFM $= \frac{1}{n-1}\sum_{j=1}^{n-1} \max_{t \in \{1,...,n-1\}}(A_{tj} - A_{nj})$. The FAA is a critical metric highlighting performance

Table 1: Overall performance (%) of CL methods across three benchmarks under various VL models.

| VL models | Method | 7 Task VG+VAW | | | 7 Task VG | | | 5 Task VAW | | |
|---|---|---|---|---|---|---|---|---|---|---|
| | | **FAA** (↑) | CAA (↑) | FFM (↓) | **FAA** (↑) | CAA (↑) | FFM (↓) | **FAA** (↑) | CAA (↑) | FFM (↓) |
| BLIP | Joint Learning | 91.90 | - | - | 95.27 | - | - | 92.60 | - | - |
| | Continual-FT [7] | 65.21 | 73.98 | 30.32 | 63.91 | 73.97 | 31.34 | 67.07 | 78.35 | 28.14 |
| | LoRA [10] | 75.39 | 76.59 | 20.73 | 69.16 | 75.89 | 28.20 | 71.54 | 79.07 | 22.48 |
| | Layered-LoRA [33] | 76.68 | 78.51 | 18.96 | 70.13 | 79.66 | 28.08 | 83.77 | 83.47 | 9.20 |
| | LwF [21] | 70.93 | 73.62 | 26.26 | 69.62 | 77.05 | 29.05 | 80.07 | 84.32 | 14.93 |
| | ZSCL [49] | 66.87 | 66.00 | 19.08 | 67.32 | 75.65 | 27.45 | 66.53 | 75.05 | 25.13 |
| | MoE-Adapters [45] | 69.90 | 74.47 | 27.11 | 64.50 | 77.18 | 34.98 | 80.09 | 83.02 | 14.36 |
| | ConStruct-VL [33] | 87.27 | 86.98 | 6.14 | 89.01 | 91.87 | 5.80 | 83.73 | 86.34 | 6.47 |
| | **ZAF (Ours)** | **90.05** | **89.45** | **3.32** | **92.49** | **92.39** | **1.97** | **89.13** | **90.03** | **3.93** |
| | *Improvement* | *2.78* | *2.47* | *2.82* | *3.48* | *0.52* | *3.83* | *5.40* | *3.69* | *2.54* |
| BLIP w/ CapFilt-L | Joint Learning | 93.72 | - | - | 95.31 | - | - | 92.90 | - | - |
| | Continual-FT [7] | 67.20 | 74.85 | 28.02 | 70.05 | 75.17 | 23.99 | 71.95 | 79.31 | 22.18 |
| | LoRA [10] | 71.97 | 76.07 | 25.27 | 69.97 | 77.52 | 28.49 | 79.66 | 82.36 | 13.78 |
| | Layered-LoRA [33] | 76.66 | 76.27 | 19.20 | 70.43 | 78.00 | 27.16 | 81.89 | 82.66 | 11.18 |
| | LwF [21] | 73.39 | 75.42 | 23.81 | 70.02 | 77.62 | 28.47 | 79.83 | 84.21 | 15.63 |
| | ZSCL [49] | 62.90 | 64.29 | 22.06 | 67.12 | 76.21 | 27.14 | 68.13 | 77.15 | 24.67 |
| | MoE-Adapters [45] | 69.76 | 73.29 | 27.34 | 63.99 | 76.19 | 35.34 | 80.01 | 84.10 | 14.43 |
| | ConStruct-VL [33] | 85.16 | 87.61 | 8.75 | 88.95 | 90.69 | 5.22 | 83.33 | 85.57 | 6.28 |
| | **ZAF (Ours)** | **89.61** | **89.65** | **4.18** | **92.53** | **92.20** | **1.72** | **89.43** | **90.20** | **3.02** |
| | *Improvement* | *4.45* | *2.04* | *4.57* | *3.58* | *1.51* | *3.50* | *6.10* | *4.63* | *3.26* |
| BLIP w/ NLVR | Joint Learning | 93.37 | - | - | 95.07 | - | - | 92.36 | - | - |
| | Continual-FT [7] | 67.23 | 73.60 | 27.96 | 73.40 | 78.60 | 20.55 | 73.19 | 80.58 | 20.69 |
| | LoRA [10] | 69.55 | 75.03 | 27.25 | 68.73 | 78.03 | 29.62 | 75.63 | 81.87 | 19.37 |
| | Layered-LoRA [33] | 80.62 | 79.89 | 13.92 | 73.03 | 81.12 | 24.99 | 83.73 | 84.26 | 9.29 |
| | LwF [21] | 73.00 | 77.26 | 23.12 | 71.11 | 79.39 | 27.09 | 82.10 | 84.69 | 11.24 |
| | ZSCL [49] | 60.27 | 67.94 | 28.48 | 65.82 | 78.06 | 27.68 | 62.03 | 74.33 | 31.20 |
| | MoE-Adapters [45] | 72.50 | 74.81 | 23.74 | 67.09 | 76.54 | 31.83 | 79.05 | 84.21 | 15.58 |
| | ConStruct-VL [33] | 85.97 | 87.00 | 6.94 | 86.96 | 90.47 | 7.91 | 84.36 | 85.93 | 5.36 |
| | **ZAF (Ours)** | **89.67** | **89.30** | **3.38** | **91.78** | **91.74** | **2.02** | **88.74** | **89.03** | **2.67** |
| | *Improvement* | *3.70* | *2.30* | *3.56* | *4.82* | *1.27* | *5.89* | *4.38* | *3.10* | *2.69* |

discrepancies between CL methods and joint learning. The CAA provides a comprehensive view of overall historical performance, and the FFM quantifies the model's capability to mitigate forgetting.

**Overall Performance:** As shown in Table 1, our ZAF consistently achieves the highest FAA, CAA, and lowest FFM, significantly outperforming competitors across three benchmarks with different pre-trained models. ZSCL [49], despite employing weight ensemble and knowledge distillation, exhibits inadequate learning on the challenging 7 Task VG+VAW and 7 Task VG benchmarks, resulting in declined performance. Similarly, MoE-Adapters, like Continual-FT [7] and LoRA [10], which lack anti-forgetting mechanisms, exhibit significant forgetting as the number of tasks increases. In contrast, both the Layered-LoRA architecture [33], with its task-specific parameter isolation strategy, and LwF [21], with only a knowledge distillation strategy, show improved performance. Notably, only ConStruct-VL [33] approaches the performance of our method but requires computationally intensive training involving adversarial attacks and utilizing all historical models for pseudo-sample generation. Despite this, ZAF shows substantial advantages, leading by **3.70**%, **4.82**%, and **4.38**% in FAA under the BLIP w/ NLVR model. Meanwhile, our method demonstrates a narrow performance gap compared to Joint Learning - the pinnacle of continual learning. The smaller this gap, the greater the challenge for CL methods to bridge it. Importantly, despite the inherent limitations in the zero-shot capabilities of some pre-trained VL models (e.g., BLIP and BLIP w/ CapFilt-L), our zero-shot antidote still effectively mitigates forgetting, underscoring its utility across a wide range of CL scenarios and pre-trained models.

**Complexity Analysis:** Table 2 presents the model size, training parameter count, and training times of various CL methods across three benchmarks. All comparisons are conducted on 7 NVIDIA GeForce RTX 3090 GPUs using 'BLIP w/NLVR', which serves as the standard configuration for subsequent experiments. As reflected in the table, ConStruct-VL exhibits exceptionally high training times, especially in the 7 Task VG+VAW scenario, attributed to the utilization of all historical models

Table 2: Comparison of training complexity among various CL methods across three benchmarks.

| Method | Model Size (M) | Train Params (M) | Train Times (h) | | |
|---|---|---|---|---|---|
| | | | 7 Task VG+VAW | 7 Task VG | 5 Task VAW |
| Continual-FT [7] | 223.94 | 223.94 | 5.57 | 2.25 | 2.90 |
| LoRA [10] | 230.13 | 6.19 | 4.64 | 1.86 | 2.82 |
| Layered-LoRA [33] | $230.13 \sim 267.29$ | 6.19 | 11.10 | 4.04 | 5.60 |
| LwF [21] | $230.13 \sim 236.33$ | 6.19 | 8.13 | 5.19 | 6.34 |
| MoE-Adapters [45] | 251.04 | 27.10 | 6.01 | 2.82 | 4.19 |
| ZSCL [49] | 223.94 | 223.94 | 11.83 | 6.44 | 7.83 |
| ConStruct-VL [33] | $230.13 \sim 267.29$ | 6.19 | 247.78 | 102.59 | 73.24 |
| ZAF (Ours) | 236.33 | 6.19 | 8.35 | 5.81 | 6.67 |
| *Training Speed Acceleration* | | $247.78/8.35 \approx 29.67$ | | $102.59/5.81 \approx 17.66$ | $73.24/6.67 \approx 10.98$ |

Table 3: Comparison of plugin performance for various CL methods across three benchmarks.

| Method | 7 Task VG+VAW | | | 7 Task VG | | | 5 Task VAW | | |
|---|---|---|---|---|---|---|---|---|---|
| | FAA (↑) | CAA (↑) | FFM (↓) | FAA (↑) | CAA (↑) | FFM (↓) | FAA (↑) | CAA (↑) | FFM (↓) |
| Joint Learning | 93.37 | - | - | 95.07 | - | - | 92.36 | - | - |
| LoRA [10] | 69.55 | 75.03 | 27.25 | 68.73 | 78.03 | 29.62 | 75.63 | 81.87 | 19.37 |
| w/ Zero-shot Antidote | **72.47** | **77.78** | **23.15** | **79.12** | **83.47** | **16.85** | **81.55** | **84.01** | **12.39** |
| Layered-LoRA [33] | 80.62 | 79.89 | 13.92 | 73.03 | 81.12 | 24.99 | 83.73 | 84.26 | 9.29 |
| w/ Zero-shot Antidote | **83.81** | **85.11** | **10.37** | **84.10** | **88.75** | **11.26** | **86.66** | **87.02** | **4.98** |
| MoE-Adapters [45] | 72.50 | 74.80 | 23.74 | 67.09 | 76.54 | 31.83 | 79.05 | 84.21 | 15.58 |
| w/ Zero-shot Antidote | **86.78** | **86.78** | **6.29** | **83.62** | **87.89** | **12.27** | **85.55** | **87.47** | **6.67** |
| ConStruct-VL [33] | 85.97 | 87.00 | 6.94 | 86.96 | 90.47 | 7.91 | 84.36 | 85.93 | 5.36 |
| w/ Zero-shot Antidote | **89.60** | **88.13** | **1.26** | **92.05** | **92.06** | **0.88** | **86.94** | **86.72** | **1.22** |
| EMA-LoRA | 77.78 | 80.96 | 17.30 | 75.02 | 82.22 | 21.55 | 83.08 | 86.21 | 10.18 |
| w/ Zero-shot Antidote (ZAF) | **89.67** | **89.30** | **3.38** | **91.78** | **91.74** | **2.02** | **88.74** | **89.03** | **2.67** |
| *Average Improvement* | *7.18* | *5.88* | *8.94* | *11.96* | *7.10* | *14.52* | *4.72* | *2.35* | *6.37* |

and complex adversarial attacks on every training sample. Although methods like MoE-Adapters and Layered-LoRA are more efficient than ConStruct-VL, their performance is still found lacking. In contrast, our method, ZAF, maintains relatively modest training times and model size, comparable to LwF - one of the simplest CL methods - yet significantly enhances anti-forgetting capabilities. Compared to ConStruct-VL, the currently best-performing method, ZAF achieves substantial training speed accelerations of **29.67**, **17.66**, and **10.98** on the three benchmarks, respectively. These results underscore ZAF's considerable advantages in terms of complexity, efficiency, and memory usage.

**Plugin Analysis:** Table 3 presents a comprehensive plugin analysis, illustrating the significant reductions in FFM and improvements in FAA and CAA achieved by our zero-shot antidote when applied to various CL methods. In practical implementations, the model from the last task of the baselines is utilized to calculate the zero-shot regularization term. The results underscore the anti-forgetting capabilities of the zero-shot antidote, as evidenced by an average reduction of 14.52% in the FFM, and increases of 11.96% in FAA and 7.10% in CAA, in the 7 Task VG benchmark. These observations further validate the universality and effectiveness of our proposed decoupled training objectives: enhancing anti-forgetting capabilities without compromising the learning process. For additional qualitative results, please refer to Appendix C.1. Furthermore, although our proposed EMA-LoRA architecture demonstrates anti-forgetting capabilities comparable to the Layered-LoRA framework, a comparative analysis when both are enhanced with the zero-shot antidote further underscores the superior performance of our approach (ZAF). This superiority arises from the synergistic effects of combining the EMA-LoRA neural architecture with zero-shot prediction stability. This configuration not only improves training efficiency but also significantly reduces memory usage. For additional results across various pre-trained models, please refer to Appendix C.2.

**Hyperparameter Analysis:** Fig. 3 showcases the performance of our ZAF method across different hyperparameter values $\alpha \in [0, 1]$, including comparative analyses with Joint Learning and our strongest competitor, ConStruct-VL. Notably, when $\alpha$ ranges from *0.65* to *0.90*, ZAF consistently outperforms ConStruct-VL by clear margins in both FAA and CAA metrics across three distinct

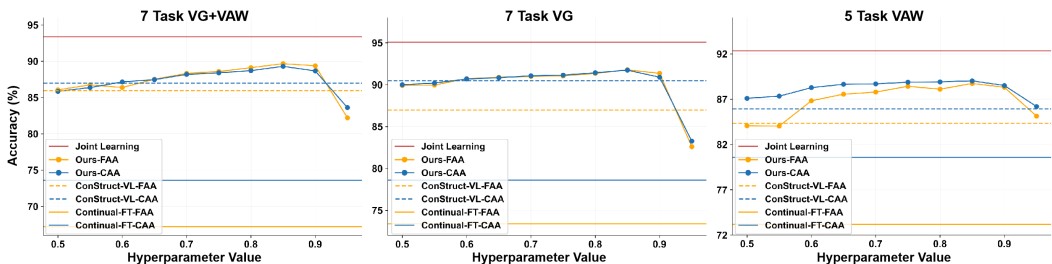

Figure 3: Comparison of FAA and CAA metrics for ZAF across various $\alpha$ values against 3 baselines.

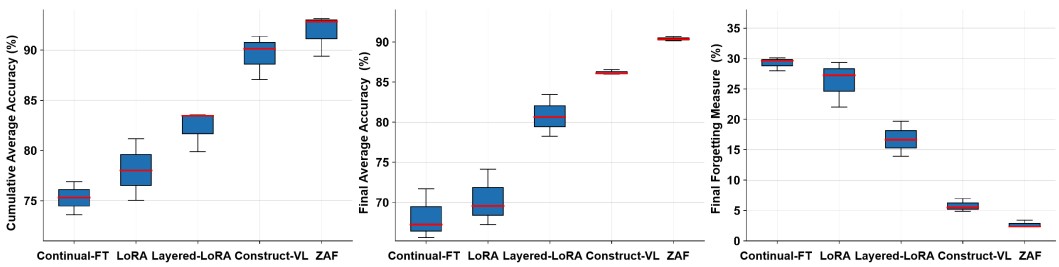

Figure 4: Results of various CL methods across three distinct task orders within 7 Task VG+VAW.

benchmarks, underscoring its superior adaptability. However, setting $\alpha$ to 0.95 leads to a notable decline in performance, reflecting the model's compromised ability to integrate new knowledge effectively. Overall, ZAF demonstrates remarkable robustness across a broad range of hyperparameter settings. Please refer to Appendix C.4 for ZAF's heatmaps across different $\alpha$ values.

**Task Order Analysis:** Fig. 4 presents the statistical results of various CL methods across three distinct task orders within the challenging 7 Task VG+VAW benchmark. Excluding the aforementioned task order within the 7 Task VG + VAW benchmark, the second and third orders are the $rel.\ spatial \rightarrow attr.\ size \rightarrow attr.\ material \rightarrow rel.\ action \rightarrow attr.\ color \rightarrow object\ state \rightarrow attr.\ action$ task sequence and the $rel.\ spatial \rightarrow attr.\ material \rightarrow attr.\ state \rightarrow attr.\ action \rightarrow attr.\ size \rightarrow rel.\ action \rightarrow attr.\ color$ task sequence, respectively. Our ZAF method exhibits remarkable robustness to variations in the task order, showing only minimal fluctuations across three performance metrics, and consistently outperforms existing CL methods by a significant margin. For detailed task sequence and quantitative results, please refer to Appendix C.3.

## 6 Discussion and Conclusion

In this work, we analyze the challenges of learning and forgetting in continual learning for pretrained VL models from both empirical and theoretical perspectives. A key finding is that zero-shot stability reliably indicates forgetting. We introduce a plug-and-play zero-shot antidote to enhance anti-forgetting capabilities across various CL methods, integrating it into our parameter-efficient EMA-LoRA architecture. This facilitates efficient adaptation and memory-free access to historical models, effectively circumventing the learning-forgetting trade-off prevalent in current CL works [3, 49, 45, 46]. Our approach demonstrates superior performance across various continual VL benchmarks and pre-trained models. We anticipate future research will explore the anti-forgetting challenge from the perspective of zero-shot prediction stability, diverging from traditional mechanisms.

This work presents several potential limitations. The effectiveness of our plug-and-play zero-shot antidote presumes the existence of a feasible CL framework, which may not be available in all contexts. Also, it is designed specifically for a sequence of downstream tasks, limiting its applicability to the original training contexts of the pre-trained models. As a fundamental research in machine learning, the potential negative societal impacts are not immediately apparent at this stage.

## Acknowledgements

This work was supported by the National Science and Technology Major Project (2023ZD0121101), and National Natural Science Foundation of China (No.62172426, 62106123).

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

# A  Theoretical Foundation

## A.1  Preliminary

For the convenience of readers who are not familiar with the PAC-Bayesian framework, we introduce the most relevant concepts from the literature here [27]. PAC-Bayesian theory studies the properties of randomized predictors, called Gibbs predictors. Formally, let $\mathcal{X}$ be an image-text input set[3], $\mathcal{Y}$ be an output set, and $\mathcal{H} \subset \{h : \mathcal{X} \to \mathcal{Y}\}$ a set of prediction functions (hypotheses). We assume that a distribution $\mathbb{D}$ is with $\mathcal{X}$, $\mathcal{Y}$, and a global labeling function $h : \mathcal{X} \to \mathcal{Y}$, where $h(x)$ generates target label for all the input, i.e., $y = h(x)$. Consider a bounded loss function $\mathcal{L} : \mathcal{Y} \times \mathcal{Y} \to [0, 1]$ for binary prediction, such that $\mathcal{L}(y_1, y_2) = 0$ holds if and only if $y_1 = y_2$. Then, we define an **expected loss** over the distribution $\mathbb{D}$ by $\mathcal{E}_{\mathbb{D}}(\mathcal{M}, h) = \mathcal{E}_{\mathbb{D}}(\mathcal{M}) = \mathbb{E}_{(x,y)\sim\mathbb{D}}[\mathcal{L}(P_{\mathcal{M}}(x), y)]$, where $P_{\mathcal{M}}(\cdot)$ is the prediction function parameterized on $\mathcal{M}$. Let $D$ denote a training set following the distribution $\mathbb{D}$ with $N$ data-label pairs. To minimize $\mathcal{E}_{\mathbb{D}}(\mathcal{M})$, we can minimize an **empirical risk** over the training set $D$ in a parameter space, i.e., $\min_{\mathcal{M}} \hat{\mathcal{E}}_D(\mathcal{M})$ where $\hat{\mathcal{E}}_D(\mathcal{M}) = \frac{1}{N} \sum_{n=1}^{N} \mathcal{L}(P_{\mathcal{M}}(x_n), y_n)$.

Below are one important definition and three critical lemmas for the proof of our Proposition 1.

**Definition 1** *(Based on Definition 1 of [1]) Given two distributions, $\mathbb{T}$ and $\mathbb{S}$, let $\mathcal{H}$ be a hypothesis class on input space $\mathcal{X}$ and denote by $I(h)$ the set for which $h \in \mathcal{H}$ is the characteristic function: that is, $x \in I(h) \Leftrightarrow h(x) = 1$. The $\mathcal{H}$-divergence between $\mathbb{T}$ and $\mathbb{S}$ is*

$$\mathrm{Div}(\mathbb{T}, \mathbb{S}) = 2 \sup_{h \in \mathcal{H}} |\mathcal{P}_{\mathbb{T}}(I(h)) - \mathcal{P}_{\mathbb{S}}(I(h))|. \tag{6}$$

**Lemma 1** *Let $\mathbb{S} = \{\mathbb{S}_i\}_{i=1}^{s}$ and $\mathbb{T}$ be $s$ source distributions and a target distribution, respectively. The $\mathcal{H}$-divergence between $\{\mathbb{S}_i\}_{i=1}^{s}$ and $\mathbb{T}$ is bounded as follows:*

$$\mathrm{Div}(\mathbb{S}, \mathbb{T}) \leq \frac{1}{\mathrm{s}} \sum_{i=1}^{s} \mathrm{Div}(\mathbb{S}_i, \mathbb{T}). \tag{7}$$

*Proof.* By the definition of $\mathcal{H}$-divergence in Definition 1,

$$\begin{aligned}
\mathrm{Div}(\mathbb{S}, \mathbb{T}) &= 2 \sup_{h \in \mathcal{H}} |\mathcal{P}_{\mathbb{S}}(I(h)) - \mathcal{P}_{\mathbb{T}}(I(h))| \\
&= 2 \sup_{h \in \mathcal{H}} \left| \sum_{i=1}^{s} \frac{1}{s} (\mathcal{P}_{\mathbb{S}_i}(I(h)) - \mathcal{P}_{\mathbb{T}}(I(h))) \right| \\
&\leq 2 \sup_{h \in \mathcal{H}} \sum_{i=1}^{s} \frac{1}{s} |\mathcal{P}_{\mathbb{S}_i}(I(h)) - \mathcal{P}_{\mathbb{T}}(I(h))| \\
&\leq 2 \sum_{i=1}^{s} \frac{1}{s} \sup_{h \in \mathcal{H}} |\mathcal{P}_{\mathbb{S}_i}(I(h)) - \mathcal{P}_{\mathbb{T}}(I(h))| \\
&= \frac{1}{s} \sum_{i=1}^{s} \mathrm{Div}(\mathbb{S}_i, \mathbb{T}),
\end{aligned} \tag{8}$$

where the first inequality is due to the triangle inequality (i.e., $|\sum_i a_i| \leq \sum_i |a_i|$) and the second inequality is by the additivity of the sup function. This finishes the proof.

**Lemma 2** *Given two distributions, $\mathbb{T}$ and $\mathbb{S}$, let $\mathcal{M}_1 \in \mathcal{H}$ and $h_1 \in \mathcal{H}$ be two prediction functions. The difference between the expected loss with $\mathbb{T}$ and $\mathbb{S}$ is bounded by the divergence between $\mathbb{T}$ and $\mathbb{S}$ as follows:*

$$|\mathcal{E}_{\mathbb{T}}(\mathcal{M}_1, h_1) - \mathcal{E}_{\mathbb{S}}(\mathcal{M}_1, h_1)| \leq \frac{1}{2} \mathrm{Div}(\mathbb{T}, \mathbb{S}), \tag{9}$$

*where $\mathrm{Div}(\mathbb{T}, \mathbb{S}) := 2 \sup_{h \in \mathcal{H}} |\mathcal{P}_{\mathbb{T}}(I(h)) - \mathcal{P}_{\mathbb{S}}(I(h))|$ is the $\mathcal{H}$-divergence for the distribution $\mathbb{T}$ and $\mathbb{S}$ ($I(h)$ is the characteristic function).*

---

[3]We abbreviate the image-text input space as $\mathcal{X}$ instead of $\{\mathcal{X}, \mathcal{I}\}$ for simplicity and ease of notation throughout this document. Likewise, $x$ instead of $\{T, I\}$.

*Proof.* By the definition of $\mathcal{H}$-divergence in Definition 1,

$$
\begin{aligned}
\mathrm{Div}(\mathbb{T}, \mathbb{S}) &= 2 \sup_{h \in \mathcal{H}} |\mathcal{P}_{\mathbb{T}}(I(h)) - \mathcal{P}_{\mathbb{S}}(I(h))| \\
&= 2 \sup_{\mathcal{M}_1, h_1 \in \mathcal{H}} \left| \mathcal{P}_{(x,y) \sim \mathbb{T}}[P_{\mathcal{M}_1}(x) \neq h_1(x)] - \mathcal{P}_{(x,y) \sim \mathbb{S}}[P_{\mathcal{M}_1}(x) \neq h_1(x)] \right| \\
&= 2 \sup_{\mathcal{M}_1, h_1 \in \mathcal{H}} \left| \mathbb{E}_{(x,y) \sim \mathbb{T}}[\mathcal{L}(P_{\mathcal{M}_1}(x), h_1(x))] - \mathbb{E}_{(x,y) \sim \mathbb{S}}[\mathcal{L}(P_{\mathcal{M}_1}(x), h_1(x))] \right| \quad (10) \\
&= 2 \sup_{\mathcal{M}_1, h_1 \in \mathcal{H}} |\mathcal{E}_{\mathbb{T}}(\mathcal{M}_1, h_1) - \mathcal{E}_{\mathbb{S}}(\mathcal{M}_1, h_1)| \\
&\geq 2|\mathcal{E}_{\mathbb{T}}(\mathcal{M}_1, h_1) - \mathcal{E}_{\mathbb{S}}(\mathcal{M}_1, h_1)|.
\end{aligned}
$$

It completes the proof.

**Lemma 3** *Let $\Theta$ be a cover of a parameter space with VC dimension d. Then, for any $\delta \in (0, 1)$ with probability at least $1 - \delta$, for any $\mathcal{M} \in \Theta$:*

$$
|\mathcal{E}_{\mathbb{D}}(\mathcal{M}) - \hat{\mathcal{E}}_D(\mathcal{M})| \leq \sqrt{\frac{d[\ln(N/d)] + \ln(1/\delta)}{2N}}, \quad (11)
$$

*where $\hat{\mathcal{E}}_D(\mathcal{M})$ is an empirical risk with N samples in its training set D.*

*Proof.* For the distribution $\mathbb{D}$, according to [22], we have

$$
\mathcal{P}(|\mathcal{E}_{\mathbb{D}}(\theta) - \hat{\mathcal{E}}_D(\theta)| \geq \epsilon) \leq 2m_{\Theta}(N) \exp(-2N\epsilon^2), \quad (12)
$$

where $m_{\Theta}(N)$ is the amount of all possible prediction results for $N$ samples, which implies the model complexity in the parameter space $\Theta$. We set $m_{\Theta}(N) = \frac{1}{2}\left(\frac{N}{d}\right)^d$ in our model, and assume a confidence bound $\epsilon = \sqrt{\frac{d[\ln(N/d)] + \ln(1/\delta)}{2N}}$. Then we get

$$
\mathcal{P}(|\mathcal{E}_{\mathbb{D}}(\theta) - \hat{\mathcal{E}}_D(\theta)| \geq \epsilon) \leq \left(\frac{N}{d}\right)^d \exp(-2N\epsilon^2) = \delta. \quad (13)
$$

Hence, the inequality $|\mathcal{E}_{\mathbb{D}}(\theta) - \hat{\mathcal{E}}_D(\theta)| \leq \epsilon$ holds with probability at least $1 - \delta$.

It completes the proof.

### A.2 Proof of Proposition 1

If we continually learn $t$ VL tasks $\mathcal{T}^1, \cdots, \mathcal{T}^t$ that follow the distribution $\mathbb{D}_1, \cdots, \mathbb{D}_t$, a solution $\mathcal{M}^t$ can be obtained with their training data $D_1, \cdots, D_t$. We define $\mathbb{D}_{1:t} := \{\mathbb{D}_1, \cdots, \mathbb{D}_t\}$ and $D_{1:t} := \{D_1, \cdots, D_t\}$. Then, for a future task $\mathcal{T}^k$ where $k \in \{t+1, \cdots, n\}$, its distribution is denoted as $\mathbb{D}_k$ with training set $D_k$. Let $\mathcal{E}_k$ be the **zero-shot generalization error** on a future task $\mathcal{T}^k$. Here, we use $\mathcal{E}_k(\mathcal{M}^t)$ instead of $\mathcal{E}_k(\mathcal{M}^t, h)$ since we assume there is no difference between labeling functions for each task for simplicity. Specifically, within the context of continual VL reasoning tasks, the labeling function $h$ is defined as a binary function. In fact, we cannot compute $\mathcal{E}_k$ since the distributions over the tasks (i.e., $\mathbb{D}_k$) and the tasks' data (i.e, $D_k$) are both unknown. However, we can approximate it by its empirical counterpart, based on the $t$ observed tasks:

$$
\begin{aligned}
\mathcal{E}_k(\mathcal{M}^t) &\leq \mathcal{E}_{1:t}(\mathcal{M}^t) + \frac{1}{2}\mathrm{Div}(\mathbb{D}_{1:t}, \mathbb{D}_k) \\
&\leq \hat{\mathcal{E}}_{1:t}(\mathcal{M}^t) + \frac{1}{2}\mathrm{Div}(\mathbb{D}_{1:t}, \mathbb{D}_k) + \sqrt{\frac{d[\ln(\bar{N}/d)] + \ln(1/\delta)}{2\bar{N}}} \\
&\leq \hat{\mathcal{E}}_{1:t}(\mathcal{M}^t) + \frac{1}{2t}\sum_{i=1}^{t}\mathrm{Div}(\mathbb{D}_i, \mathbb{D}_k) + \sqrt{\frac{d[\ln(\bar{N}/d)] + \ln(1/\delta)}{2\bar{N}}},
\end{aligned} \quad (14)
$$

where $\hat{\mathcal{E}}_{D_{1:t}}(\mathcal{M}^t) = \frac{1}{t}\sum_{i=1}^{t}\hat{\mathcal{E}}_{D_i}(\mathcal{M}^t)$ because $||\mathcal{M}^{i+1} - \mathcal{M}^i||_2 \leq \Delta$ that represents the weight vectors for continual tasks are only minor variations. The three inequalities hold due to Lemma 2, Lemma 3 and Lemma 1, respectively. $\mathbb{D}_{1:t} := \{\mathbb{D}_i\}_{i=1}^t$ and we rewrite a mixture of all the $t$ distributions as $\mathbb{D}_{1:t} := \frac{1}{t}\sum_{i=1}^{t}\mathbb{D}_i$ using convex combination. $\bar{N} = \left(\frac{1}{t}\sum_{i=1}^{t}\frac{1}{N_i}\right)^{-1}$ is the harmonic

mean of the training example sizes (i.e., $N_i$) in each task. Of note, $\text{Div}(\mathbb{D}_i, \mathbb{D}_k)$ is rewritten as $\text{Div}(\mathcal{T}_i, \mathcal{T}_k)$ in the main body to avoid using the distribution notation $\mathbb{D}_i$ and $\mathbb{D}_k$.

Furthermore, for previously learned tasks $\mathcal{T}^s$ where $s \in \{1, \cdots, t-1\}$, let $\mathcal{E}_s$ denote the **generalization error on old task** $\mathcal{T}^s$. We cannot compute $\mathcal{E}_s$ since the distributions over the old tasks (i.e., $\mathbb{D}_s$) and the tasks' data (i.e, $D_s$) are both unavailable at current time $t$. However, we can approximate it by its empirical counterpart, based on $t$ observed tasks:

$$
\begin{aligned}
\mathcal{E}_s(\mathcal{M}^t) &\leq \mathcal{E}_{1:t}(\mathcal{M}^t) + \frac{1}{2}\text{Div}(\mathbb{D}_{1:t}, \mathbb{D}_s) \\
&\leq \hat{\mathcal{E}}_{1:t}(\mathcal{M}^t) + \frac{1}{2}\text{Div}(\mathbb{D}_{1:t}, \mathbb{D}_s) + \sqrt{\frac{d[\ln(\bar{N}/d)] + \ln(1/\delta)}{2\bar{N}}} \\
&\leq \hat{\mathcal{E}}_{1:t}(\mathcal{M}^t) + \frac{1}{2t}\sum_{i=1}^{t}\text{Div}(\mathbb{D}_i, \mathbb{D}_s) + \sqrt{\frac{d[\ln(\bar{N}/d)] + \ln(1/\delta)}{2\bar{N}}},
\end{aligned}
\tag{15}
$$

where $\text{Div}(\mathbb{D}_i, \mathbb{D}_s)$ is rewritten as $\text{Div}(\mathcal{T}_i, \mathcal{T}_s)$ in the main body to avoid using the distribution notation $\mathbb{D}_i$ and $\mathbb{D}_s$.

Interestingly, the bound in both Eq. (14) and Eq. (15) contains two types of complexity terms that correspond to two levels of our model: $\text{Div}(\mathbb{D}_{1:t}, \cdot)$ corresponds specifically to the similarity between target task with all $t$ observed tasks, while $\sqrt{\frac{d[\ln(\bar{N}/d)] + \ln(1/\delta)}{2\bar{N}}}$ belongs to the similarity among all tasks in general.

Based on the definition of $\mathcal{H}$-divergence in Definition 1, for future task $\mathcal{T}^k$ and $k \in \{t+1, \cdots, n\}$, we have

$$
\begin{aligned}
\sum_{i=1}^{t}\text{Div}(\mathbb{D}_i, \mathbb{D}_k) &= 2\sum_{i=1}^{t}\sup_{h \in \mathcal{H}}|\mathcal{P}_{\mathbb{D}_i}(I(h)) - \mathcal{P}_{\mathbb{D}_k}(I(h))| \\
&\geq 2\sum_{i=1}^{t}\sup_{h \in \mathcal{H}}|\mathcal{P}_{\mathbb{D}_i}(I(h)) - \mathcal{P}_{\mathbb{D}_s}(I(h))| \\
&= \sum_{i=1}^{t}\text{Div}(\mathbb{D}_i, \mathbb{D}_s),
\end{aligned}
\tag{16}
$$

where the inequality holds due to $s \in \{1, \cdots, t-1\}$, and current model $\mathcal{M}_t$ incorporates knowledge of the distribution from the old task $\mathcal{T}^s$. In fact, $\text{Div}(\mathbb{D}_{1:t}, \mathbb{D}_k)$ can also be interpreted as the extent to which model $\mathcal{M}_t$ needs to be adjusted to learn task $\mathcal{T}^k$. Obviously, $\text{Div}(\mathbb{D}_{1:t}, \mathbb{D}_k)$ is generally larger than $\text{Div}(\mathbb{D}_{1:t}, \mathbb{D}_s)$ because $\mathcal{T}^s$ has already been learned previously, even though its data and model are no longer available.

It can be concluded from the two bounds in Eq. (14) and Eq. (15) that a model $\mathcal{M}^t$ that has learned $t$ tasks demonstrates consistent reasoning abilities on both future tasks and previously learned tasks. This consistency hinges on the model's ability to zero-shot prediction and anti-forgetting. Additionally, the ability to maintain statistical consistency, i.e., showing stable performance across different tasks, is a key indicator of the success of a continual learning strategy, especially to the anti-forgetting challenge. In practice, ensuring this consistency of trend requires carefully designed learning strategies and appropriate regularization measures to enhance anti-forgetting capabilities without interfering with the learning process.

## B  Implementation Details on Baselines

In the implementation of LwF [21], we conduct it based on LoRA, instead of fine-tuning the entire model. For each incremental task, we preserve the LoRA from the last task for the distillation procedure. Regarding ConStruct-VL, the results are reproduced using the official codes and hyperparameters. We use update steps $n_{adv} = 10$, the step size $\lambda_{adv} = 0.01$, and the loss weight $\rho = 0.2$. For ZSCL [49], which employs a KD loss to align with the original CLIP model, the hyperparameter $I$ dictates the weight ensemble frequency, as specified in the original papers. The KD weight is set to 1, and the hyperparameter $I$ is set to 5 to enhance learning capability and overall performance, although it still performs inadequately in the challenging 7 Task VG+VAW benchmark. For MoE-Adapters [45], originally proposed for image classification tasks requiring task-id for inference, we

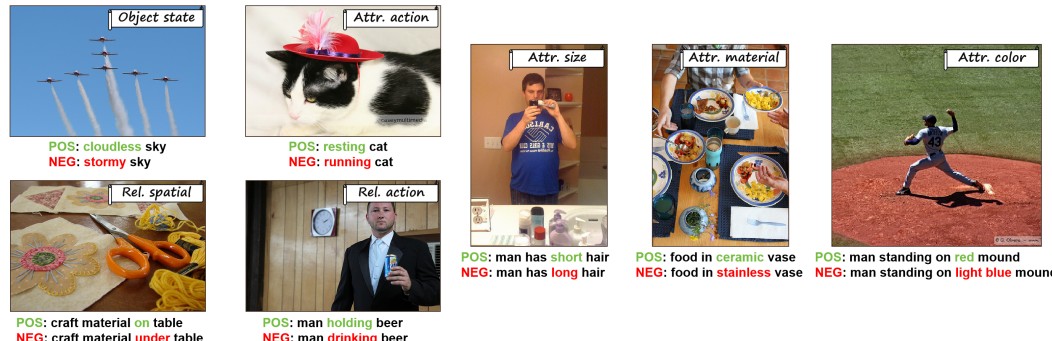

Figure 5: Examples of Structured VL Concept reasoning task.

adapt its implementation to include a single task-dependent router for our VL task where there is no task boundary. Following the original paper, the number of experts is set to 14, and the router chooses the top-4 experts during training and inference.

# C  Extended Results

In this section, we provide some extended results for the main text.

## C.1  Extended Plugin Analysis

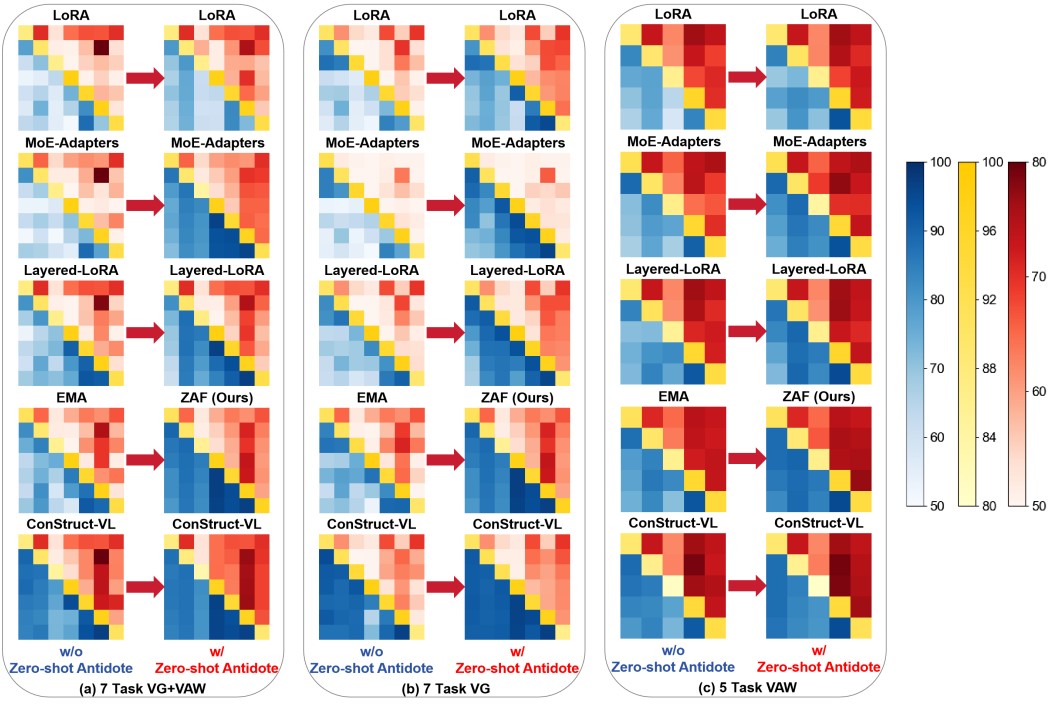

Figure 6: Effect of our zero-shot antidote on various CL methods across three benchmarks.

We present a comprehensive plugin analysis by detailing the performances of various methods with and without our zero-shot antidote in Table 3. Here, we offer visual comparisons in Fig. 6 to illustrate the impact of our zero-shot antidote on enhancing both zero-shot prediction and anti-forgetting capabilities across different methods. As demonstrated in the figure, incorporation our zero-shot antidote significantly improves zero-shot prediction performance on unseen tasks (indicated by the upper triangular red area) for each method without compromising their learning abilities (represented

Table 4: Component analysis of our ZAF under three pre-trained VL models.

| VL models | Method | 7 Task VG+VAW | | | 7 Task VG | | | 5 Task VAW | | |
|---|---|---|---|---|---|---|---|---|---|---|
| | | **FAA** (↑) | CAA (↑) | FFM (↓) | **FAA** (↑) | CAA (↑) | FFM (↓) | **FAA** (↑) | CAA (↑) | FFM (↓) |
| BLIP | Joint Learning | 91.90 | - | - | 95.27 | - | - | 92.60 | - | - |
| | EMA-LoRA | 76.24 | 79.89 | 20.11 | 64.50 | 78.96 | 32.63 | 81.92 | 86.28 | 12.69 |
| | ZAF (Ours) | **90.05** | **89.45** | **3.32** | **92.49** | **92.39** | **1.97** | **89.13** | **90.03** | **3.93** |
| | *Improvement* | *13.81* | *9.56* | *16.79* | *27.99* | *13.43* | *30.66* | *7.21* | *3.75* | *8.76* |
| BLIP w/ CapFilt-L | Joint Learning | 93.72 | - | - | 95.31 | - | - | 92.90 | - | - |
| | EMA-LoRA | 75.80 | 79.56 | 20.66 | 73.77 | 82.04 | 23.54 | 80.74 | 85.87 | 14.08 |
| | ZAF (Ours) | **89.61** | **89.65** | **4.18** | **92.53** | **92.20** | **1.72** | **89.43** | **90.20** | **3.02** |
| | *Improvement* | *13.81* | *10.09* | *16.48* | *18.76* | *10.16* | *21.82* | *8.69* | *4.33* | *11.06* |
| BLIP w/ NLVR | Joint Learning | 93.37 | - | - | 95.07 | - | - | 92.36 | - | - |
| | EMA-LoRA | 77.78 | 80.96 | 17.30 | 75.02 | 82.22 | 21.55 | 83.08 | 86.21 | 10.18 |
| | ZAF (Ours) | **89.67** | **89.30** | **3.38** | **91.78** | **91.74** | **2.02** | **88.74** | **89.03** | **2.67** |
| | *Improvement* | *11.89* | *8.34* | *13.92* | *16.76* | *9.52* | *19.53* | *5.66* | *2.82* | *7.51* |

by the diagonal yellow area). Moreover, the anti-forgetting capability (shown in the lower triangular blue area) of each method is also substantially improved. Although ConStruct-VL already exhibits strong anti-forgetting capabilities, its performance is further enhanced when its zero-shot prediction ability is augmented with our antidote. These visual comparisons not only support our empirical findings that the stability of consecutive zero-shot predictions can serve as a reliable indicator of its anti-forgetting capabilities but also verify that our zero-shot antidote successfully decouples learning from forgetting.

## C.2 Detailed Component Analysis

In the main text, we present part results for EMA-LoRA architecture and ZAF, noting that EMA-LoRA demonstrates performance comparable to Layered-LoRA, and the integration of the zero-shot antidote significantly enhances its performance. Here, we perform an extended ablation study in Table 4 to delve into the interaction between the EMA-LoRA architecture and zero-shot antidote under three pre-trained VL models. The excellent performance improvement in the table can be attributed to the close cooperation between the EMA-LoRA architecture and the zero-shot antidote. As illustrated in Fig. 2 (b), on the one hand, the EMA-LoRA architecture effectively ensembles historical knowledge to provide informative zero-shot supervision. On the other hand, the zero-shot antidote offers an effective solution that separates the learning processes from the problem of anti-forgetting. By incorporating them, ZAF significantly mitigates forgetting and provides overwhelming performance compared to existing baselines, along with significantly improved efficiency.

## C.3 Detailed Performance Analysis across Different Task Orders

Table 5: Overall performance (%) of various CL methods across different task orders.

| Method | Task Order 1 | | | Task Order 2 | | | Task Order 3 | | |
|---|---|---|---|---|---|---|---|---|---|
| | **FAA** (↑) | CAA (↑) | FFM (↓) | **FAA** (↑) | CAA (↑) | FFM (↓) | **FAA** (↑) | CAA (↑) | FFM (↓) |
| Joint Learning | 93.37 | - | - | 93.37 | - | - | 93.37 | - | - |
| Continual-FT [7] | 67.23 | 73.60 | 27.96 | 71.68 | 75.31 | 22.24 | 65.61 | 76.88 | 29.65 |
| LoRA [10] | 69.55 | 75.03 | 27.25 | 74.13 | 81.17 | 21.99 | 67.21 | 78.00 | 29.38 |
| Layered-LoRA [33] | 80.62 | 79.89 | 13.92 | 75.92 | 83.47 | 19.66 | 78.23 | 76.96 | 24.86 |
| LwF [21] | 73.00 | 77.26 | 23.12 | 75.38 | 79.59 | 20.58 | 74.76 | 81.32 | 21.21 |
| ZSCL [49] | 60.27 | 67.94 | 28.48 | 60.27 | 67.94 | 28.48 | 55.46 | 70.95 | 35.68 |
| MoE-Adapters [45] | 72.50 | 74.81 | 23.74 | 71.42 | 76.96 | 24.86 | 72.61 | 78.73 | 23.27 |
| ConStruct-VL [33] | 85.97 | 87.00 | 6.94 | 86.07 | 91.36 | 5.49 | 86.56 | 90.15 | 4.88 |
| ZAF (Ours) | **89.67** | **89.30** | **3.38** | **90.67** | **93.12** | **2.33** | **90.16** | **92.84** | **2.35** |
| *Improvement* | *3.70* | *2.30* | *3.56* | *4.60* | *1.76* | *3.16* | *3.60* | *2.69* | *2.53* |

In the main text, Fig. 4 illustrates the results of various CL methods across three different task orders, demonstrating the robustness of our approach. In Table 5, we further present the quantitative

performance of all these methods. It is evident that our ZAF method consistently outperforms all baselines by substantial margins. Moreover, our method exhibits minimal performance variation across different task orders, with the FAA consistently hovering around 90%. In contrast, none of the baselines come close to reaching the FAA near 90%.

## C.4 Extended Hyperparameter Analysis

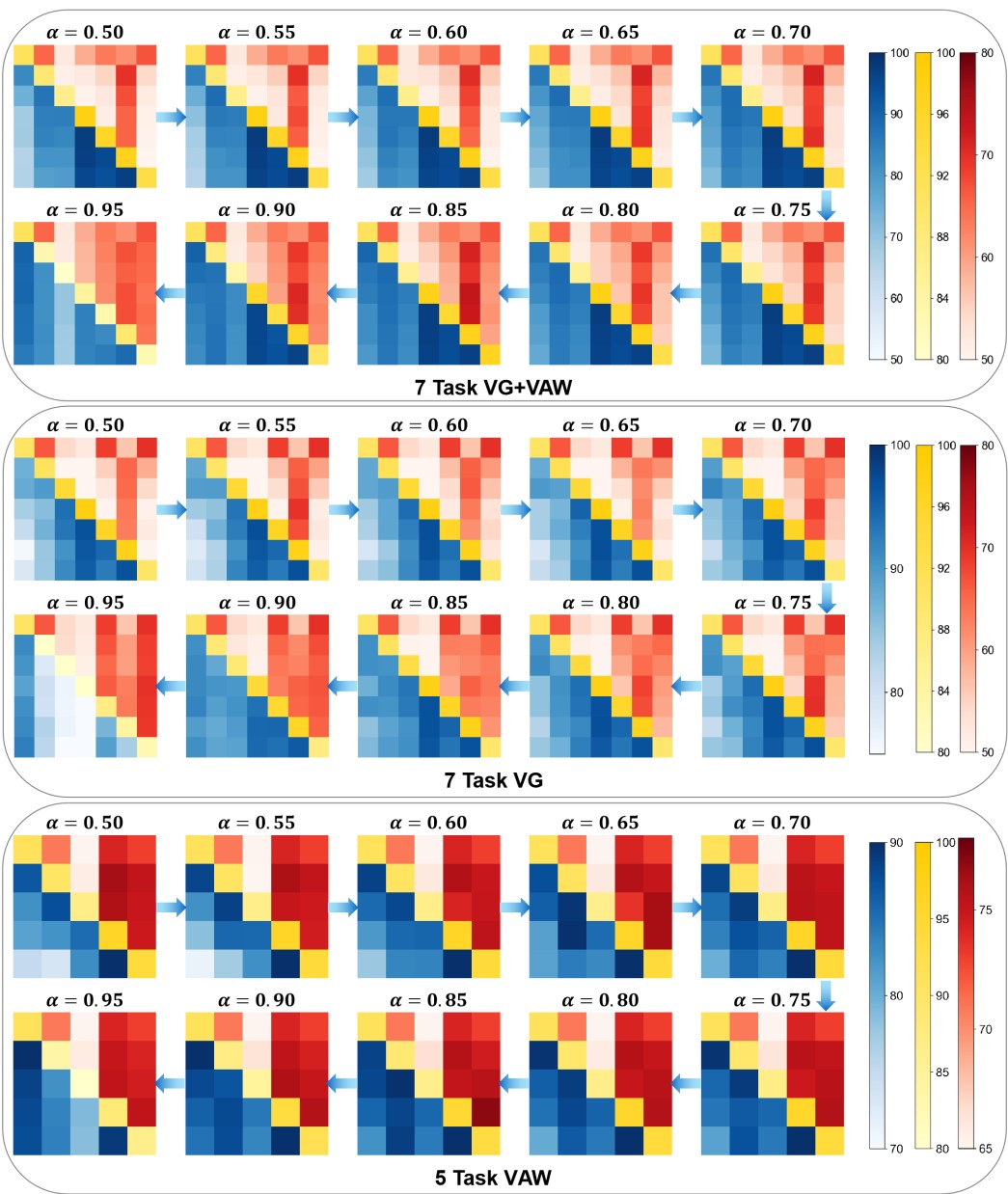

Figure 7: Heatmap changes for our ZAF across various $\alpha$ values in three benchmarks.

The hyperparameter $\alpha$ introduced by our method is important to control the rate of parameter updates. As depicted in Fig. 3, our ZAF method exhibits remarkable robustness across a wide range of $\alpha$ values compared to three baselines. Further insights into the qualitative impact of $\alpha$ on the anti-forgetting capability are provided in Fig. 7, which presents heatmaps of our ZAF performance across various $\alpha$ values. It can be observed from these visualizations that an increase in the $\alpha$ value significantly

improves the zero-shot prediction stability, thereby enhancing the model's anti-forgetting capability. This enhancement is attributed to preserving more historical knowledge at higher $\alpha$ values, which ensures more stable zero-shot predictions and less forgetting. However, it is important to note that setting $\alpha$ to 0.9 and 0.95 compromises the model's ability to integrate new knowledge effectively, resulting in reduced learning ability (the brighter diagonal yellow area) and a decline in overall performance. These observations underscore the effectiveness of our ZAF implementation and, in turn, also validate our empirical findings.

## C.5 Wild Data Composition Analysis

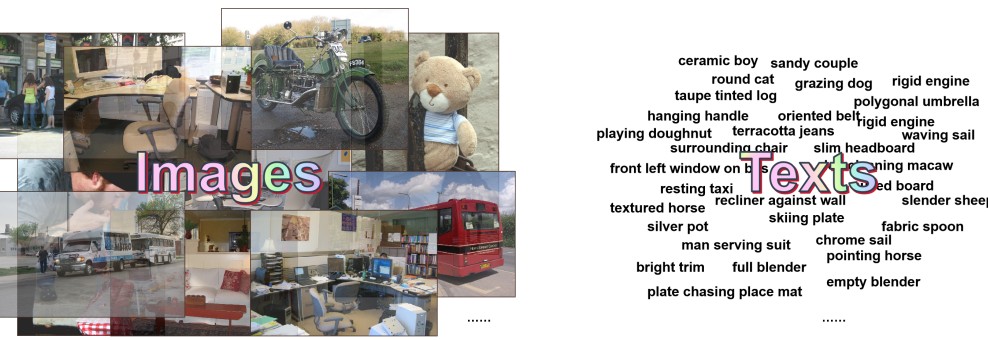

Figure 8: Randomly selected examples from our wild dataset.

The construction of the wild dataset is important for stabilizing zero-shot predictions across a broad spectrum of downstream tasks. Like the SVLC multimodal benchmark, each sample in our wild dataset consists of triplets, including a single image and two texts. These texts do not correspond to the image content; one text is manipulated by altering a specific concept found in the other. To ensure that images and texts remain unpaired, we shuffle them in the training batch, as done in the provided code. For our dataset, we first collect a substantial collection of images from the VG and VAW datasets. We then identify the main objects within these images to form a list of nouns. Next, we use ChatGPT4 [25] to generate a list of concepts words covering a wide range of structured VL concepts, such as shape (e.g., round, square, triangular), texture (e.g., smooth, coarse, sandy), and brightness (e.g., faint, light, bright). For each image, we then randomly combine words from the noun and concept lists in a grammatical format to create one text. We create another text for the image by replacing the concept word in this text with another randomly selected concept from our concept list. It is important to note that the clarity of these texts (e.g., round cat, sandy couple, faint bus) is not prioritized, as the main objective is to evaluate the model's zero-shot ability to handle image-text pairs. The generation process allows us to create wild data efficiently with the help of ChatGPT4. In total, our dataset comprises *12,358* unique images and *30,144* unique texts, resulting in *21,006 triplets*. Fig. 8 shows some randomly selected examples from our wild dataset.

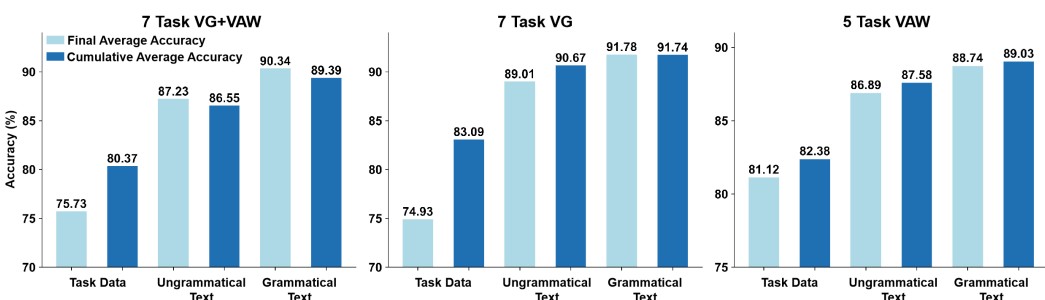

Figure 9: Performance of our ZAF using the wild data with different compositions.

As shown in Fig. 9, our comparative analysis evaluates the performance of ZAF using different compositions of wild data. First, employing only current task data to calculate the zero-shot loss

yields notably limited results. This limitation stems from coupled training targets, similar to traditional knowledge distillation techniques, which rely on the same data source, as illustrated in Subsection 4.2. Moreover, utilizing wild data with ungrammatical text leads to decreased performance, likely because the model tends to overlook image information and classify image-text pairs with ungrammatical texts as inherently negative. This limitation also hampers the effective evaluation of the model's zero-shot capabilities. In contrast, employing grammatically correct text achieves the best performance in our experiments. The figure clearly demonstrates that employing grammatically correct texts significantly enhances both FAA and CAA, thereby confirming the effectiveness of our process for creating wild data.

## C.6  Extended Comparisons

Table 6: Extended comparison with unimodal CL methods across three benchmarks.

| Method | 7 Task VG+VAW | | 7 Task VG | | 4 Task VAW | |
|---|---|---|---|---|---|---|
| | **FAA** (↑) | CAA (↑) | **FAA** (↑) | CAA (↑) | **FAA** (↑) | CAA (↑) |
| Joint Leaning | 93.37 | - | 95.07 | - | 92.26 | - |
| L2 [34] | 76.36 | 85.59 | 77.40 | 87.12 | 83.72 | 86.01 |
| EWC [15] | 73.29 | 81.81 | 72.78 | 82.61 | 86.00 | 87.26 |
| L2P [43] | 66.96 | 76.46 | 60.88 | 69.23 | 62.27 | 68.68 |
| Dual Prompt [42] | 58.59 | 68.64 | 65.55 | 66.94 | 60.07 | 71.98 |
| ConStruct-VL [33] | 85.40 | 90.88 | 86.99 | 93.00 | 87.50 | 89.77 |
| ZAF (Ours) | **90.77** | **93.23** | **91.71** | **94.73** | **89.26** | **90.95** |
| *Average Improvement* | *5.37* | *2.35* | *4.72* | *1.73* | *1.76* | *1.18* |

In Table 1, we provide the overall performance of our ZAF method and other multimodal baselines. To conduct more comprehensive comparisons with CL methods, Table 6 presents the performances for widely-used regularization-based unimodal CL methods such as L2 [34] and Elastic Weight Consolidation (EWC) [15], alongside prompt-based baselines like Learning to Prompt (L2P) [43] and Dual Prompt (DP) [42]. Here, we use the $rel.\ spatial \rightarrow attr.\ size \rightarrow attr.\ material \rightarrow rel.\ action \rightarrow attr.\ color \rightarrow object\ state \rightarrow attr.\ action$ 7-task sequence in the '7 Task VG+VAW' benchmark and '7 Task VG' benchmark, and the $attr.\ action \rightarrow object\ state \rightarrow attr.\ color \rightarrow attr.\ size$ 4-task sequence in the '4 Task VAW' benchmark, using the 'BLIP w/ CapFilt-L' model. The task sequence settings and results are excerpted from [33].

The table shows a substantial performance gap between unimodal and multimodal methods, highlighting the insufficient adaptation ability of visual CL methods in multimodal scenarios. Additionally, our ZAF method consistently outperforms ConStruct-VL with notable margins in the FAA metric across three benchmarks, further proving its effectiveness. The more challenging the benchmark, the more remarkable the margins become.

