# OpenReview forum: "Stabilizing Zero-Shot Prediction: A Novel Antidote to Forgetting in Continual Vision-Language Tasks"
_NeurIPS.cc/2024/Conference — NeurIPS 2024 poster_

### Official Review · Reviewer_wYS7 · 2024-07-09

**Soundness:** 3
**Presentation:** 4
**Contribution:** 3
**Rating:** 6
**Confidence:** 4

**Summary:**

The paper introduces a novel continual learning (CL) method called ZAF (Zero-shot Antidote to Forgetting) designed to enhance the retention of previously learned skills in vision-language (VL) models without replaying historical data. The proposed approach leverages zero-shot stability as an indicator of anti-forgetting capabilities, incorporating a stability regularization term and an efficient EMA-LoRA neural architecture.

**Strengths:**

1. The introduction of zero-shot stability regularization and the EMA-LoRA architecture provides a novel way to tackle the forgetting problem in CL.

2. Empirical and Theoretical Rigor: The combination of empirical studies and theoretical analysis strengthens the validity of the proposed method.

3. Demonstrates significant improvements over existing methods across multiple benchmarks.

**Weaknesses:**

1. The paper evaluates the performance of ZAF using a relatively small number of subtasks. Conducting experiments with a larger variety of subtasks or across more extensive datasets could provide a more comprehensive validation of the method's effectiveness.

2. Alongside the incremental learning performance on downstream tasks, it would be beneficial to report the model's zero-shot performance across a broad range of domains.

**Questions:**

How much wild data this method used in experiments.

**Limitations:**

As a fundamental research in machine learning, the potential negative societal impacts are not immediately apparent at this stage.

---

> ### Author Rebuttal · Authors · 2024-08-07
>
> Thank you for your thoughtful summary and for acknowledging the contributions of our work with the ZAF model. We greatly appreciate your recognition of the paper's clear presentation, significant performance improvements, and its empirical and theoretical rigor. Your appreciation for the introduction of zero-shot stability regularization and the EMA-LoRA architecture is also highly valued by our team.
>
> 1. **Conducting experiments with a larger variety of subtasks or across more extensive datasets**
>
> In response to your suggestion, we expanded our evaluation framework to include two additional benchmarks, enhancing the diversity of our experiments. These benchmarks, namely the '5 Task VG+VAW' and the '5 Task VAW + 2 Task VG,' encompass a variety of distinct concepts from both the VG (Visual Genome) and VAW (Visual Attributes in the Wild) datasets. Below are the comparative results illustrating the performance of ConStruct-VL and our ZAF model:
>
> [5 Task VG+VAW Benchmark:]
> | Method  | 'BLIP' | 'BLIP w/ CapFilt-L' | 'BLIP w/ NVLR' |
> |-------|----------------|-----------|------------|
> |       | FAA (↑)  CAA (↑)  FFM (↓) | FAA (↑)  CAA (↑)  FFM (↓) | FAA (↑)  CAA (↑)  FFM (↓)
> |ConStruct-VL |84.02 86.38 6.17 | 82.69 85.43 6.61 | 83.63 84.97 5.01
> |ZAF| 89.26 90.05 3.61 | 89.28 89.91 3.39 | 89.05 89.10 2.40
> |**Improvement**| 5.24 3.67 2.56 | 6.59 4.48 3.22 | 5.42 4.13 2.61|
>
> [5 Task VAW + 2 Task VG Benchmark:]
> | Method  | 'BLIP' | 'BLIP w/ CapFilt-L' | 'BLIP w/ NVLR' |
> |-------|----------------|-----------|------------|
> |       | FAA (↑)  CAA (↑)  FFM (↓) | FAA (↑)  CAA (↑)  FFM (↓) | FAA (↑)  CAA (↑)  FFM (↓)
> |ConStruct-VL|84.11 86.69 6.45|83.13 85.97 6.94| 83.73 85.43 7.13
> |ZAF|90.13 90.96 2.12 | 89.87 90.63 2.97| 90.04 90.92 2.38
> |**Improvement**|6.02 4.27 4.33|6.74 4.66 3.97| 6.31 5.49 4.75
>
> These results affirm that ZAF not only sustains exceptional anti-forgetting performance across various task configurations but also consistently surpasses ConStruct-VL across all metrics. The improvements in FAA, CAA, and FFM are particularly notable, showcasing significant gains. This comprehensive evaluation substantiates ZAF's robustness and versatility, confirming its superior performance in more complex and varied learning environments.
>
> 2. **Report the model's zero-shot performance across a broad range of domains**
>
> In response to your suggestion, we have expanded our evaluation to include not only the continual learning performance but also the zero-shot performance of our ZAF model during the continual learning process. The results are tabulated as follows:
>
> [7 Task VG+VAW benchmark using 'BLIP':]
> |ZAF|$T_1$|$T_2$|$T_3$|$T_4$|$T_5$|$T_6$|$T_7$|AVG
> |---|---|---|---|---|---|---|---|---|
> |zero-shot performance|46.32|62.34|51.55|50.86|54.64|64.80|51.84|54.62
> |continual learning performance|90.59|90.91|85.94|97.36|95.64|97.34|92.50|92.90
>
> [7 Task VG+VAW benchmark using 'BLIP w/ NVLR':]
> |ZAF|$T_1$|$T_2$|$T_3$|$T_4$|$T_5$|$T_6$|$T_7$|AVG
> |---|---|---|---|---|---|---|---|---|
> |zero-shot performance|66.40|65.31|51.49|53.95|58.10|72.52|59.69|61.07
> |continual learning performance|91.03|89.70|85.57|97.36|95.09|97.08|92.17|92.57
>
> [7 Task VG benchmark using 'BLIP':]
> |ZAF|$T_1$|$T_2$|$T_3$|$T_4$|$T_5$|$T_6$|$T_7$|AVG
> |---|---|---|---|---|---|---|---|---|
> |zero-shot performance|47.00|55.91|50.00|50.03|50.54|60.60|50.16|52.03
> |continual learning performance|93.37|90.64|93.71|98.04|96.07|97.56|89.91|94.19
>
> [7 Task VG benchmark using 'BLIP w/ NVLR':]
> |ZAF|$T_1$|$T_2$|$T_3$|$T_4$|$T_5$|$T_6$|$T_7$|AVG
> |---|---|---|---|---|---|---|---|---|
> |zero-shot performance|68.00|66.50|50.51|50.16|59.11|63.27|61.06|59.80
> |continual learning performance|90.31|90.64|93.03|97.74|96.07|97.38|89.40|93.51
>
> [5 Task VAW benchmark using 'BLIP':]
> |ZAF|$T_1$|$T_2$|$T_3$|$T_4$|$T_5$|AVG
> |---|---|---|---|---|---|---|
> |zero-shot performance|46.08|71.72|58.42|67.86|65.39|61.89
> |continual learning performance|92.12|92.13|87.06|96.00|94.07|92.28
>
> [5 Task VAW benchmark using 'BLIP w/ NVLR':]
> |ZAF|$T_1$|$T_2$|$T_3$|$T_4$|$T_5$|AVG
> |---|---|---|---|---|---|---|
> |zero-shot performance|66.39|70.85|66.48|75.13|76.98|71.17
> |continual learning performance|91.07|88.63|86.32|94.94|93.44|90.88
>
> As observed, our ZAF model delivers satisfactory continual learning performance across a sequence of downstream tasks. However, its absolute zero-shot performance across a broad range of unseen domains remains constrained, despite achieving the zero-shot stability we aim for. This outcome is expected, as utilizing current task data naturally bolsters continual learning performance at the expense of zero-shot prediction capabilities. Inspired by your feedback, we recognize the essential need to simultaneously optimize both continual learning and zero-shot learning capabilities to extend the practical applications of our model. Addressing this dual optimization challenge is the primary focus of our ongoing projects. We aim to significantly improve the model’s adaptability and generalization across unfamiliar tasks and domains, ensuring robust performance in a wider array of settings.
>
> 3. **How much wild data this method used in experiments.**
>
> As mentioned in Line 194, further details about the construction of the wild dataset are provided in Appendix C.5. In total, our wild dataset comprises 12,358 unique images and 30,144 texts, resulting in 21,006 unique triplets. Importantly, these images/texts are entirely disjoint from both the original pretraining and downstream data, ensuring that our dataset provides a unique and independent testing environment. The wild dataset has been included in the supplementary materials accompanying our manuscript.

---

> ### Comment · Reviewer_wYS7 · 2024-08-09
>
> The author has fully answered my question. I maintain a positive evaluation of this paper.

---

> ### Author Response · Authors · 2024-08-11
> **Response to Reviewer Comments**
>
> Thank you once again for your valuable feedback and positive evaluation. Your insights have been instrumental in enhancing our work.

---

### Official Review · Reviewer_DvN8 · 2024-07-12

**Soundness:** 3
**Presentation:** 2
**Contribution:** 3
**Rating:** 5
**Confidence:** 5

**Summary:**

This paper tackles continual pretraining / training for vision-language models by introducing both exponentially moving averaged LoRA, and more importantly, replay during training on additional unaligned text and image data. In doing so, the authors show that higher performance on standard continual VL tasks can be achieved. To motivate and support this approach, the authors also conduct a simple zero-shot generalization versus forgetting study, and highlight the connection between retained zero-shot generalization and reduced forgetting, using it as a (theoretically motivated) driver for the regularization / replay on additional data.

**Strengths:**

* The paper is sufficiently well written to make it easy to follow.

* The performance of both EMA-LoRA and particular ZAF compared to other approaches is, on a purely numerical basis, very convincing and significant.

**Weaknesses:**

__[1]__ On the sensibility of zero-shot prediction as indicator for forgetting.

Section 3.2 has effectively no actionability as insights are entirely of qualitative nature. It’s crucial to provide detailed numerical comparisons (L149-151). Moreover, it is quite expected that reduced changes in forward generalization / generalization on yet unseen data come with reduced forgetting, since it likely indicates that the model has forgotten loss / has experienced less catastrophic feature overwriting. For a model that zero-shot generalizes sufficiently well to all tasks before adaptation, retention of this zero-shot performance on remaining tasks naturally results in better performance on tasks encountered during training.

> This has been explored before in e.g. Stojanovski et al. 2022 (https://arxiv.org/abs/2211.03186), which already show that an EMA-style objective helps when deploying pretrained models in a continual scenario.

In my eyes, the reasoning starts from the wrong premise - it is the choice of a more suitable method that results in higher zero-shot and seen-task / anti-forgetting performance. As a result, it is somewhat meaningless to use these metrics as a regularizer for a method, since these are the exact metrics methods want to optimize for anyway (e.g. also L187 “underscores that the model’s capabilities in zero-shot prediction can reliably indicate its anti-forgetting capabilities”).

> However, I’d love to hear the authors perspective on this. How would the methods zero-shot performance cause the “anti-forgetting” capabilities - as opposed to a confounder (i.e. the choice of method) improving both?

---

__[2]__ Method and Experiments

__2.1__ I would love to get more details on the motivation behind the loss term in Eq. 4.

> What exactly is happening with each loss term, and what is the exact motivation behind these choices?

__2.2__ More importantly however, if I understand correctly, the authors simply introduce additional data during tuning compared to existing approaches. Even if unlabelled, this is effectively semi-supervised training, which is a strict extension over supervised training on a same-sized supervised set. However, the authors do not discuss any semi-supervised learning works (nor do they compare against it).

> Firstly, I might have missed it, but where is the additional data taken from? Secondly, how is the proposed approach not just replay on additional data similar to the original pretraining data - something that is an orthogonal improvement for any replay-free baseline method compared against?

__2.3__ I’m not sure if the authors are claiming EMA training as a contribution (e.g. for EMA-LoRA).

> If so, it would be great if the authors could relate this to existing works such as Stojanovski et al. 2022 linked above.

**Questions:**

I am currently advocating for rejection for reasons listed above. In particular, I currently fail to see how improvements don't just stem from conducting semi-supervised training and effectively introducing a replay buffer on data similar to that encountered during original pretraining, which gives an expected, yet unfair advantage to existing methods (and which is not an actual contribution unfortunately).

I'm willing to raise my score, but would need to see this, as well as the smaller issues listed above, addressed.

**Limitations:**

The paper explicitly discusses limitations and societal impact.

---

> ### Author Rebuttal · Authors · 2024-08-07
>
> Thank you for your insightful review and for recognizing the clarity and performance improvements of our manuscript. We have addressed the common questions separately and will now respond to the specific concerns you raised.
>
> **1. The relation of zero-shot prediction stability, absolute zero-shot performance, anti-forgetting, and model choice**
>
> - In our study, we concentrate on preserving acquired knowledge during the continual learning process by **stabilizing** the zero-shot prediction capabilities of a model as it learns. This stabilization is achieved **independently of any pre-existing generalization capabilities of a pre-trained model across tasks before adaptation**. Notably, the zero-shot data used are completely disjoint from both the original pretraining data and the downstream task data, ensuring a rigorous test of the model's adaptability and generalization capabilities.
>
> - Our primary concern is the **relative change** in zero-shot performance throughout the continual learning process, rather than the absolute zero-shot performance on downstream tasks. The rationale behind this approach includes:
>   - Even if a pre-trained model (such as BLIP) does not initially exhibit strong zero-shot abilities, it can still effectively preserve acquired knowledge by stabilizing zero-shot predictions.
>   - Conversely, a pre-trained model (e.g., BLIP w/ NLVR) with better initial zero-shot ability may still experience significant forgetting if zero-shot prediction stability is not maintained.
>
> The detailed results are presented in the table below.
>
> |BLIP|7 Task VG+VAW | 7 Task VG | 5 Task VAW |
> |-|-|-|-
> |Zero-shot Accuracy|50.74|50.83|50.42
> |Final Forgetting w/o $L_{ZS}$|20.11|32.63|12.69
> |Final Forgetting w/ $L_{ZS}$|3.32|1.97|3.93
>
> |BLIP w/ CapFilt-L|7 Task VG+VAW | 7 Task VG | 5 Task VAW |
> |-|-|-|-
> |Zero-shot Accuracy|49.60|50.88|49.23
> |Final Forgetting w/o $L_{ZS}$|20.66|23.54|14.08
> |Final Forgetting w/ $L_{ZS}$|4.18|1.72|3.02
>
> |BLIP w/ NLVR|7 Task VG+VAW | 7 Task VG | 5 Task VAW |
> |-|-|-|-
> |Zero-shot Accuracy|67.89|68.82|70.39
> |Final Forgetting w/o $L_{ZS}$|17.30|21.55|10.18
> |Final Forgetting w/ $L_{ZS}$|3.38|2.02|2.67
>
> - It's crucial to understand that **zero-shot performance and anti-forgetting performance are not inherently linked** metrics in continual learning methods. Specifically, high zero-shot performance does not necessarily equate to enhanced seen-task or anti-forgetting performance. As detailed in Proposition 1, the generalization errors for both seen and unseen tasks are bounded by the discrepancy between task distributions (see term $Div$ and L185-187). This means that for continual tasks that closely resemble each other, such as classification tasks using CLIP noted in Stojanovski et al. 2022, there is a consistent correlation between high zero-shot and anti-forgetting performance when unseen tasks closely align with continual tasks. However, this correlation does not extend to dissimilar tasks, such as the reasoning tasks in our studies utilizing BLIP.
>
> **2. Where is the additional/wild data taken from?**
>
> As mentioned in L194, further details about the construction of the wild dataset are provided in **Appendix C.5**. Importantly, these images/texts are entirely disjoint from both the original pretraining and downstream data, ensuring that our dataset provides a unique and independent testing environment. The wild dataset has been included in the supplementary materials. Additionally, Figure 9 presents a performance comparison of our method using the wild data with various compositions, further demonstrating its robustness.
>
> **3. Clarify the fairness of comparisons with existing methods**
>
> - As noted in L12, the proposed zero-shot stability regularization, implemented by the inclusion of additional data, facilitates the preservation of acquired knowledge **in a plug-and-play manner**. To ensure **fair comparisons**, we have also conducted **ablation studies** on existing continual learning methods with our zero-shot regularization and wild data, as detailed **in L297-311 and Table 3**. It can be observed that with the implementation of our zero-shot stability regularization, all existing methods consistently demonstrate significant improvements in their anti-forgetting capabilities, as indicated by reduced forgetting measurement (FFM).
>
> - Furthermore, as shown in Table 3, even though the robust baseline ConStruct-VL is augmented with a replay buffer for old samples, when enhanced with zero-shot stability regularization, our ZAF method still achieves comparable or slightly superior results.
>
> **4: Discuss some semi-supervised learning(SSL) works**
>
> Our method aligns with the inductive SSL framework, which utilizes both labeled and unlabeled data for training, as described in the survey by Van Engelen et al. [3]. However, our approach incorporates several distinct features:
>
> - **Purpose**: Unlike conventional SSL that enhances generalization within the labeled data space, our method uses unlabeled data to improve anti-forgetting capabilities across all previously learned tasks.
>
> - **Assumptions**: Our model doesn't rely on typical SSL assumptions like the manifold and low-density separation assumptions. Instead, it handles non-stationary task distributions, a common scenario in continual learning.
>
> - **Training Strategy**: Our approach diverges from standard SSL strategies like self-training or co-training by implementing stability regularization on predictions of unlabeled data, which aids in maintaining consistent task performance.
>
> - **Empirical Insights**: Empirical results (see Figure 9) demonstrate that leveraging unlabeled data from a diverse range of distributions is more effective than restricting to the labeled task distribution, challenging a core principle of traditional SSL where similarity between labeled and unlabeled data is preferable.

---

> > ### Comment · Reviewer_DvN8 · 2024-08-09
> > **Re: Rebuttal**
> >
> > I thank the authors for the detailed feedback, clarififying some misunderstandings on my end, and discussing the relation to Stojanovski et al. 2022.
> >
> > However, I am still not convinced that comparisons made to other methods are fair, and what the actual contributions are that the reader should extract from this paper.
> >
> > ---
> >
> > For one, e.g. Stojanovski et al. 2022 utilize EMA for CL during training, which is different to what the authors note in the shared response (_"Our approach differentiates from traditional EMA-style methods[1,2] by incorporating EMA-LoRA not only during inference but also throughout training"_).
> >
> > ---
> >
> > Moreover, while the proposed method or model may not build on the same intuitions such as standard SSL mehods (_"manifold and low-density separation"_ as per the authors), it does not mean that the overall setup is not conducting semi-supervised learning as training happens on both supervised and unsupervised data (in contrast to other methods compared to). It doesn't matter if certain particular SSL approaches such as self-training aren't utilized - as soon as joint training over supervised and _additional_ unsupervised data occurs, one falls within the SSL regime. This is particularly problematic, as all the other reference methods only utilize the labelled training data.
> >
> > Similarly, the authors argue that the results _"challenge a core principle of traditional SSL where similarity between labeled and unlabeled data is preferable"_ - it would be great if the authors could provide references for this claim for completeness.
> >
> > More importantly however, the authors explicit note in their own rebuttal that their continual learning setting differs from standard CL, in that _"correlation does not extend to dissimilar tasks, such as the reasoning tasks in our studies utilizing BLIP"_. Indeed, the authors evaluate on much more complex and mixed natural image domains, in which cases including data from a diverse set of domains should actually help (as it mimics in nature the data used for pretraining). This is seen in the consistent improvements across methods in Tab. 3 when including additional data.
> >
> > ---
> >
> > Put together, in my eyes, the authors get performance gains by adding meaningful (w.r.t. to the evaluated benchmarks) additional unsupervised data, which has limited novelty, and makes e.g. Tab. 1 meaningless. At the same time, exponential moving averages for methods in continual learning have already been explored before (see [1,2] in shared response).
> > As such, it would be great if the authors could very precisely state what the relevant technical novelty is. I'm still happy to raise my score, however need this aspect to very clearly carved out.

---

> > > ### Author Response · Authors · 2024-08-12
> > > **Rebuttal by Authors**
> > >
> > > Thank you for your thoughtful response. Below we will further address your concerns.
> > >
> > > 1.**Our EMA-LoRA and existing EMA-style methods**
> > >
> > > - **Training**:
> > >   - [1,2] adapt solely to the target distribution by $\min_{\theta_{fast}}L(f_{\theta_{fast}}(x),y)$, where $\theta_{fast}$ is updated through backpropagation. The EMA model $\theta_{slow}$ is updated from $\theta_{fast}$ via interpolation but doesn't partake in loss calculation, primarily serving for inference. For instance, [2] uses $\theta^{on}$ as $\theta_{fast}$ and $\theta^{off}$ as $\theta_{slow}$.
> > >   - Our approach enhances this by integrating the EMA model into the full training cycle. Here $\mathcal{W}$ and $\widehat{\mathcal{W}}$ correspond to $\theta_{fast}$ and $\theta_{slow}$. We redefine the optimization as $\min_{\mathcal{W}}L(f_{\mathcal{W}}(x),y)+L_{ZS}(f_{\mathcal{W}}(x^{wild}),f_{\widehat{\mathcal{W}}}(x^{wild}))$, **making $\widehat{\mathcal{W}}$ actively contribute to the loss calculations and updates**.
> > >
> > > - **Inference**:[1,2] use EMA model $\theta_{slow}$ for inference. In contrast, our model enhances inference flexibility by allowing the use of either the EMA model $\widehat{\mathcal{W}}$ (ZAF) or the current training model $\mathcal{W}$ (ZAF_variant), ensuring strong performance regardless of the model selected.
> > >
> > > |BLIPw/CapFilt-L|7TaskVG+VAW|7TaskVG|5TaskVAW
> > > |-|-|-|-
> > > |Method|FAA CAA FFM |
> > > |ConStruct-VL|85.16 87.61 8.75|88.95 90.69 5.22|83.33 85.57 6.28
> > > |ZAF_variant|89.25 88.52 4.80|90.85 89.96 3.39|88.53 89.02 3.89
> > > |ZAF|89.61 89.65 4.18|92.53 92.20 1.72|89.43 90.20 3.02
> > >
> > > 2.**Relation to SSL**
> > >
> > > - A core principle of traditional SSL is that labeled and unlabeled data should come from similar distributions, with deviations potentially leading to performance issues, as clearly noted in [3,4].
> > >
> > > - Contrarily, our findings(Fig.9) show that leveraging a diverse range of unlabeled data distributions can be more effective, challenging the conventional SSL assumption that emphasizes similarity between labeled and unlabeled data.
> > >
> > > - The classification as SSL depends on **why and how unlabeled data is used**. We use it to bolster anti-forgetting across all CL tasks, not just to enhance performance on current tasks. This is implemented through a novel regularization strategy, diverging from standard SSL methods like self-training.
> > >
> > > - Our method is akin to **self-taught learning**[5,6], where the focus is on using unlabeled data to develop a versatile feature extractor, not constrained by class labels or distributions matching those of labeled data.
> > >
> > > [3]Safe deep semi-supervised learning for unseen-class unlabeled data
> > >
> > > [4]They are Not Completely Useless:Towards...
> > >
> > > [5]Self-taught learning:transfer learning from unlabeled data
> > >
> > > [6]Robust and Discriminative Self-Taught Learning
> > >
> > > 3.**Performance gains not solely attributable to unlabeled data**
> > >
> > > - **Simply adding more unlabeled data does not guarantee improved CL performance**. The table below demonstrates that using self-supervised contrastive consistency regularization(SSL) instead of zero-shot stability regularization with the same wild data results in a notable performance decline.
> > >
> > > |BLIP w/ NLVR|7TaskVG+VAW|7TaskVG|5TaskVAW
> > > |-|-|-|-
> > > |Method|FAA CAA FFM||
> > > |ConStruct-VL|85.97 87.00 6.94|86.96 90.47 7.91|84.36 85.93 5.36
> > > |SSL|80.54 83.20 8.23|77.87 78.85 16.98|82.24 83.14 9.42
> > > |ZAF|89.67 89.30 3.38|91.78 91.74 2.02|88.74 89.03 2.67
> > >
> > > - Fig.9 illustrates that **our approach to constructing unlabeled data is flexible and robust**, specifically designed to encompass a broad spectrum of future CL tasks. Importantly, this data is entirely distinct from both the original pretraining and downstream datasets, thereby **upholding CL principles**.
> > >
> > > - The concept of **leveraging unlabeled data in CL is not new and has been previously explored**. For instance, [7] discusses the use of data from different datasets to simulate the data distribution of old tasks.
> > >
> > > - Table 1 presents a Joint Learning upper bound performance, ConStruct-VL that utilizes **data replay for old tasks**, setting a challenging benchmark, and **ZSCL that employs the same wild data** but with a different methodology.
> > >
> > > [7]Overcoming Catastrophic Forgetting with Unlabeled Data in the Wild
> > >
> > > 4.**Technical novelty**
> > >
> > > - A significant innovation is the **empirical and theoretical demonstration** that zero-shot prediction stability, when adapting to new tasks, effectively indicates anti-forgetting capabilities. This is rigorously supported by theoretical proofs.
> > >
> > > - We develop **a plug-and-play zero-shot stability regularization**, readily integrable into various frameworks to combat forgetting.
> > >
> > > - Our **EMA-LoRA architecture** enhances stabilization efficiency by integrating the EMA mechanism throughout the **training process**, boosting the CL performance.
> > >
> > > - Similar to self-taught learning, we **utilize a large volume of easily accessible unlabeled data** but focus distinctly on anti-forgetting, thereby **decoupling** it from new task learning.

---

> > > > ### Comment · Reviewer_DvN8 · 2024-08-13
> > > > **Response to Response**
> > > >
> > > > I think the authors again for the detailed reply, which has helped clarify my issues regarding the comparison to related works [1,2].
> > > >
> > > > While I am still not convinced w.r.t. the separation from SSL, the newely provided table does provide good experimental support that the proposed zero-shot stability regularization indeed helps to better leverage additional unlabeled data for continual learning.
> > > >
> > > > Given that these two points, alongside my previously raised questions have been consequently mostly addressed, I am raising my score and am leaning towards acceptance.

---

> > > > > ### Author Response · Authors · 2024-08-13
> > > > > **Response to Reviewer Comments**
> > > > >
> > > > > Thank you for your thoughtful review and for patiently accepting our clarifications. Your feedback has greatly helped us rethink our work and improve our paper. We truly appreciate your time and effort.

---

### Official Review · Reviewer_CbcE · 2024-07-15

**Soundness:** 3
**Presentation:** 3
**Contribution:** 3
**Rating:** 5
**Confidence:** 4

**Summary:**

The paper presents a novel continual learning (CL) method named ZAF (Zero-shot Antidote to Forgetting) aimed at enhancing the performance of pre-trained vision-language (VL) models in zero-shot prediction tasks. The authors identify zero-shot stability as a key indicator of a model’s ability to retain previously learned information and propose a zero-shot stability regularization technique to maintain this stability. ZAF employs an EMA-LoRA architecture for parameter-efficient learning and applies zero-shot antidote regularization to wild data, decoupling the processes of learning and forgetting.

**Strengths:**

1.The introduction of zero-shot stability as a measure for anti-forgetting is a novel and promising concept. This approach allows for a plug-and-play solution that can be integrated with existing CL frameworks.

2.The use of an Exponential Moving Average (EMA) combined with Low-Rank Adaptation (LoRA) for parameter efficiency is a significant contribution. This architecture balances the need for efficient adaptation to new tasks while maintaining access to historical models.

3.The extensive experiments conducted on various VL benchmarks demonstrate the effectiveness of ZAF. ZAF’s approach to training significantly reduces computational costs compared to methods like ConStruct-VL.

**Weaknesses:**

1.	For the empirical study in Section 3.2 and Figure 1, the evaluations is the same as in [49] except for the datasets used, are there any differences? Additionally, could you provide a more detailed discussion on why “a model’s stability in zero-shot predictions can reflect its anti-forgetting capabilities” (line 152) can be observed in Figure 1?
2.    The proposed method is restrictede to specific pre-trained model (BLIP). Can it be generalzied to more general multi-modal models, such as CLIP?
3.	For the Loss_ZS, have you performed ablations using other measurements such as cross-entropy, which aligns with the basic Loss_CE optimization of the model, or using KL divergence, etc.?
4.	The idea of using EMA for updating offline LoRA has been proposed in a previous work, LAE [1]. Could you discuss the differences between the two approaches?
5.	The experiment results in Table 1 are lower than those presented in ConStruct-VL [33], and even the proposed method shows lower results compared to [33]. Considering the experiment settings are identical to those in [33], could you explain the reasons behind this discrepancy?
6.	Additionally, could you clarify why, in some cases, the FAA is actually higher than the CAA? Which mechanism in your method allows predictions for previous tasks to improve when learning a new task?

[1] Gao, Q., Zhao, C., Sun, Y., Xi, T., Zhang, G., Ghanem, B., & Zhang, J. (2023). A unified continual learning framework with general parameter-efficient tuning. ICCV 2023.

I will reconsider my rating upon the rebuttal.

**Questions:**

please see weakness.

**Limitations:**

The authors discussed some limitations in the supplementary.

---

> ### Author Rebuttal · Authors · 2024-08-07
>
> Thank you for your supportive review and valuable suggestions. We have addressed the common questions separately and will now respond to the unique concerns raised.
>
> 1. **Differences between the evaluations in our Figure 1 and ZSCL[49]**
>
> Our approach diverges from ZSCL in both objectives and evaluation metrics. While ZSCL aims to optimize zero-shot and continual learning performances simultaneously, our research focuses on enhancing continual learning through zero-shot stability regularization. Evaluation Metrics Comparison:
> - **ZSCL**:
>   - Avg Metric: Average accuracy across all tasks and timestamps.
>   - Last Metric: Performance of all tasks after the full learning sequence.
>   - Transfer Metric: Zero-shot transfer ability, calculated from the upper-right matrix.
> - **Ours**:
>   - Final Average Accuracy(FAA): Similar to ZSCL's Last metric.
>   - Cumulative Average Accuracy(CAA): Measures average accuracy across learned tasks, derived from the lower-left matrix.
>   - Final Forgetting Measure(FFM): Assesses knowledge retention derived from the lower-left matrix.
>
> We also introduce a **zero-shot stability metric (ZSS in Table R1)**, which evaluates the consistency of zero-shot performance across learning stages, computed in the upper-right triangle but **totally differnt from  ZSCL's Transfer Metric**. These tailored metrics underscore our specific focus on continual learning's unique challenges, demonstrating a clear methodological divergence from ZSCL's broader objectives.
>
> 2. **Generalzied to more general multi-modal models, such as CLIP**
>
> We evaluated our method using the BLIP model and its two variants. Here, we tested CLIP, a predecessor of BLIP known for its robust performance. Below is the performance comparison across three datasets:
>
> |Methods|7 Task VG+VAW |7 Task VG|5 Task VAW|
> |-|-|-|-|
> ||FAA (↑) CAA (↑) FFM (↓)|FAA (↑) CAA (↑) FFM (↓)| FAA (↑) CAA (↑) FFM (↓)
> |LoRA|67.79 73.31 24.96|63.48 76.30 36.57|75.22 74.08 15.68
> |EMA-LoRA|69.24 73.16 22.76|64.35 77.34 33.87|76.25 76.61 14.41
> |ZAF|75.04 75.79 16.80|68.57 78.223 28.77|77.64 77.05 13.30
>
> Our findings show that CLIP, though versatile, **underperforms compared to BLIP in our tests**. BLIP's optimized representational capabilities for image and text inputs make it better suited for the complex reasoning tasks we evaluate. This highlights the importance of **selecting the right pre-trained model for specific experimental challenges**.
>
> 3. **Ablations using other measurements such as cross-entropy(CE) or KL divergence for the $\mathcal{L}{ZS}$**
>
> Thank you for the suggestion. Initially, we used the L1-norm for the $\mathcal{L}_{ZS}$ loss term due to its computational efficiency. To further test our method's robustness, we experimented with CE and KL divergence as alternative loss measures. Below is a summary of the performance metrics for these loss variants across three datasets:
>
> |Model|Metric|7 Task VG+VAW |7 Task VG|5 Task VAW|
> |-|-|-|-|-|
> |||FAA (↑) CAA (↑) FFM (↓)|FAA (↑) CAA (↑) FFM (↓)|FAA (↑) CAA (↑) FFM (↓)
> |BLIP|$\mathcal{l}_1$ norm|90.05  89.45  3.32|92.49 92.39 1.97|89.13 90.03 3.93
> ||$KL$/$CE$|89.32 88.44 3.55|90.64 90.92 2.97|88.14 88.70 3.97
> |BLIP w/ NLVR|$\mathcal{l}_1$ norm|89.67 89.30 3.38|91.78 91.74 2.02 |88.74 89.03 2.67
> ||$KL$/$CE$|89.46 88.37 2.99|90.64 90.02 2.97|88.25 87.81 2.38
>
> Performance is comparable across all three loss metrics, with a slight edge for L1-norm, showing **the robustness of our method to variations in loss design**. Notably, KL and CE produced identical results, reflecting their mathematical equivalence in binary prediction tasks, where predictions fit Bernoulli distributions.
>
> 4. **Reasons for lower experimental results in Table 1 compared to ConStruct-VL[33]**
>
> Discrepancies between our results and those reported for ConStruct-VL[33] in Table 1 arise from **different task orders** in our continual learning experiments. As detailed in Appendix C.3, **Task Order 2**, which aligns with ConStruct-VL's sequence, involves a reduced number of training samples per task. This effectively lowers the complexity and challenge of the task environment, leading to higher performance metrics:
>
> |Method|FAA (↑)|CAA (↑)|FFM (↓)
> |-|-|-|-|
> |Reported ConStruct-VL in[33]|85.4|90.88|-
> |Reprodeced ConStruct-VL|86.07|91.36|5.49
> |ZAF|90.67|93.12|2.33
>
> The forgetting metric used by ConStruct-VL[33] averages forgetting across all tasks, different from our methodology, making direct comparisons somewhat skewed. Nevertheless, our data confirms that ZAF **not only replicates but significantly outperforms ConStruct-VL** under comparable conditions, demonstrating our approach's robustness and superior efficacy.
>
> 5. **Why the FAA is higher than the CAA in some cases**
>
> There is **no preset rule** that FAA should be higher or lower than CAA; this depends on the model's learning dynamics and how well it manages forgetting. For example, in our ZAF model using the 'BLIP w/ NVLR' framework on the 7-task VG+VAW benchmark, FAA occasionally exceeds CAA when newer tasks are learned effectively or earlier tasks are well-retained:
>
> |ZAF|$T_1$|$T_2$|$T_3$|$T_4$|$T_5$|$T_6$|$T_7$|AVG
> |-|-|-|-|-|-|-|-|-|
> |$T_1$|91.03|-|-|-|-|-|-|91.03
> |$T_2$|89.63|89.70|-|-|-|-|-|89.66
> |$T_3$|86.15|88.59|85.57|-|-|-|-|86.77
> |$T_4$|85.10|87.85|85.15|97.36|-|-|-|88.87
> |$T_5$|81.97|87.57|83.86|96.95|95.09|-|-|89.09
> |$T_6$|81.97|86.83|82.76|96.96|94.37|97.08|-|90.00
> |$T_7$|79.79|86.27|82.11|96.29|94.20|96.87|92.17|89.67
>
> Here, the FAA (89.67) slightly exceeds the CAA (89.30, the average of the AVG column) due to strong performance on the last task and minimal forgetting of earlier tasks.

---

> ### Comment · Reviewer_CbcE · 2024-08-12
>
> I appreciate the authors' response. It addresses part of my concerns but also raises some new concerns (along with checking other reviewers' comments).
> 1. Thanks for highlighting the detailed differences between the analysis in ZSCL. Although different specific metrics are used, the overall analysis methodology is still very similar to ZSCL.
> 2. I can see that the results of using BLIP are better than CLIP. It is not surprising. I can also understand it is easy to obtain good performance using a strong baseline or pre-trained model, and it is not only about performance. It is mainly for checking whether the proposed techniques are general and robust to obtain good performance in different conditions.
>
> I am still concerned (and even more concerned) about this related to the baseline model, experimental setting, and the loss functions.
>
> 1) How were the experiments conducted with CLIP? Is the wild reference dataset still used? Is it also used for the experiments with CLIP?
>
> 2) How much performance gain (with BLIP or CLIP) is from the additionally introduced reference dataset? It is also one of the concerns of reviewer DvN8.
>
> 3) Can the proposed (rest) techniques work without the additional wild reference dataset, i.e., without loss L_ZS?
>
> 4) I am concerned that the proposed methods (the whole method) and the performances are highly tight with the selected baseline model BLIP and the specifically used unsupervised wild training data.
>
> Since the continual learning research is mainly about how to maintain performance and avoid forgetting, heavily relying on the specifically trained model and additional data diverges from the original purposes and is slightly misleading, although it can improve performances on the benchmarks. Considering some empirical results are still valuable, I give the score 5. I am still open to reconsidering my score upon the response.

---

> > ### Author Response · Authors · 2024-08-13
> >
> > Thank you for your response. We appreciate the opportunity to address your concerns.
> >
> > 1.**Experiments with CLIP**
> >
> > - The CLIP model, designed primarily for image-text similarity tasks, is not optimal for reasoning tasks requiring a classification head. In our experiments with three datasets featuring triplets (one image and two texts—one positive and one negative), we **adhered to CLIP's inference by evaluating similarities and assigning labels based on which text is more similar to the image**. For training, **consistent with CLIP's standard methodology**, we treated paired image-text as positive and unpaired ones as negative. However, CLIP's reliance on comparative similarity limits its ability to perform inference on a single image-text pair. Therefore, we chose BLIP and its variants for reasoning tasks, as they demonstrated superior performance.
> >
> > - To develop LoRA, EMA-LoRA, and ZAF, we integrate low-rank adapters to CLIP's encoders. Specifically, in ZAF, we employed wild data alongside our zero-shot regularization, represented by $\mathcal{L}_{\text{ZS}}(S^t(T\_\text{wild}, I\_\text{wild}), \widehat{S^t}(T\_\text{wild}, I\_\text{wild}))$, where $S^t(\cdot)$ and $\widehat{S^t}(\cdot)$ denote the image-text similarities from the current and EMA-based models, respectively.
> >
> > - Our ZAF model shows performance gains even when implemented with CLIP. As noted in the previous response, ZAF **clearly outperforms baseline methods when using CLIP**, demonstrating the effectiveness of our zero-shot regularization. Although the improvements are less pronounced than with BLIP, this is expected given CLIP's focus on image-text similarity rather than reasoning tasks. This distinction highlights the adaptability of our method across different pre-trained models.
> >
> > 2.**Performance gains not solely from wild data**
> >
> > - Foundation: A key innovation of our study is the **empirical and theoretical demonstration** that zero-shot prediction stability when adapting to new tasks, effectively signals anti-forgetting capabilities. This finding underpins our use of wild data and the associated zero-shot stability regularization. Without wild data, the model would simply rely on the basic EMA-LoRA architecture, lacking innovative regularization.
> >
> > - Methodology: **Simply adding more unlabeled data does not guarantee improved CL performance**. The table below demonstrates that using the same wild data under a self-supervised contrastive consistency loss(SSL), as opposed to zero-shot stability regularization, shows a notable performance decline.
> >
> > |Method|7TaskVG+VAW|7TaskVG|5TaskVAW
> > |-|-|-|-
> > ||FAA CAA FFM
> > |ConStruct-VL|85.97 87.00 6.94|86.96 90.47 7.91|84.36 85.93 5.36
> > |SSL|80.54 83.20 8.23|77.87 78.85 16.98|82.24 83.14 9.42
> > |ZAF|89.67 89.30 3.38|91.78 91.74 2.02|88.74 89.03 2.67
> >
> > - Precedents: The use of additional unlabeled data in CL is an established concept and has been previously explored[1,2]. Typically, these methods selectively use data from external datasets to mimic the distribution of old tasks to combat forgetting, less competitive compared to our method **in both learning and anti-forgetting**. The ZSCL model in Table 1, despite using wild data, underperforms our ZAF method, highlighting the effectiveness of our regularization.
> >
> > These points articulate our distinct work of using unlabeled data not merely as an additive resource but as a strategic element integral to enhancing CL performance while addressing the limitations of traditional methods.
> >
> > [1]Overcoming Catastrophic Forgetting with Unlabeled Data in the Wild.
> >
> > [2]Learning to imagine: Diversify memory for...
> >
> >
> > 3.**ZAF method without loss $L_{ZS}$**
> >
> > When our ZAF is applied without wild data and $L_{ZS}$, it effectively becomes the EMA-LoRA. As shown in Table 3, even in this reduced form, it surpasses architectures such as LoRA and MoE-Adapters.
> >
> > 4.**reliability to additional data**
> >
> > - Fig.9 shows that our way of constructing unlabeled data is flexible and robust. Our ZAF outperforms ConStruct-VL, which relies on replaying old data, even when the texts in wild data are ungrammatical(87.23 and 85.97). We adhere to two primary rules for constructing wild data: it covers a broad spectrum of distributions and remains completely distinct from the original pretraining and downstream datasets. This strategy ensures compliance with CL principles without access to old or test data.
> >
> > - Before submitting our work, we tested ZAF's performance using various compositions of wild data on the 7TaskVG+VAW benchmark. The notation ZAF$^{x}$ represents adding ${x}$% new data whose concept is not from the downstream benchmark to our wild data. The results below demonstrate the robustness of ZAF and its zero-shot loss, showing that performance gains are not dependent on specifically tailored data.
> >
> > |Method|FAA CAA FFM
> > |-|-
> > |ConStruct-VL|85.97 87.00 6.94
> > |ZAF|89.67 89.30 3.38
> > |ZAF$^{15}$|89.90 89.21 3.02
> > |ZAF$^{25}$|88.97 88.35 3.50
> > |ZAF$^{50}$|88.60 88.03 4.22
> > |ZAF$^{75}$|89.25 88.67 3.85

---

> > ### Author Response · Authors · 2024-08-13
> >
> > Dear Reviewer CbcE,
> >
> > As the discussion period is set to end in one day, we wanted to kindly remind you that we have posted our latest response addressing your concerns. We have also addressed similar concerns raised by Reviewer DvN8. If you have any further questions or require clarification on any points, please do not hesitate to reach out. We are committed to addressing your queries promptly.
> >
> > We greatly value your feedback and look forward to your insights.
> >
> > Best,
> >
> > Authors

---

### Author Rebuttal · Authors · 2024-08-07

Thank you to all reviewers for your insightful feedback. We will first address common concerns about our zero-shot stability and EMA-LoRA architecture, followed by detailed responses to each reviewer’s specific comments.

**Q1: How zero-shot prediction stability indicates anti-forgetting capabilities**

**A1**: To elucidate how zero-shot stability serves as an indicator of anti-forgetting, we will explore this relationship from both empirical and theoretical perspectives. Supporting details are provided in Table R1 and Figure R1 of the newly submitted PDF file.
- **Empirical Evidence**: We included numerical values for all heatmap entries of Figure 1 within Figure R1. These metrics illustrate: Zero-shot Stability assessed through the ZSS metric in the performance matrix's upper-right triangle (red area); Anti-forgetting Ability evaluated using the FFM metric in the lower-left triangle (blue area); and Learning New Tasks measured by the average values along the diagonal (yellow area). These values show that higher zero-shot stability corresponds to improved anti-forgetting performance, demonstrating a direct correlation without negatively impacting new task learning abilities. Here are comparative metrics:
|7 Task VG+VAW|Zero-shot stability|Anti-forgetting|Learning|/|7 Task VG|Zero-shot stability|Anti-forgetting|Learning|
|-|-|-|-|-|-|-|-|-
|MoE-Adapters|5.21|23.74|92.85|/|MoE-Adapters|1.57|31.83|94.37|
|LoRA|6.03|27.25|92.91|/|LoRA|3.83|29.62|93.79|
|Layred-LoRA|4.95|13.92|92.83|/|Layred-LoRA|3.42|24.99|93.92|
|ConStruct-VL|3.39|6.94|92.16|/|ConStruct-VL|2.70|7.91|93.43|

- **Theoretical Support**: Per Proposition 1, the anti-forgetting ability ($\mathcal{E}\_{s}(\mathcal{M}^{t}) - \mathcal{E}\_{s}(\mathcal{M}^{s})  \leq Bound1$) and the stability of zero-shot predictions ($|\mathcal{E}{k} (\mathcal{M}^{t}) - \mathcal{E}{k} (\mathcal{M}^{s})|  \leq Bound2$) are tightly linked. The bounds provided in Table R1 show that the stability in zero-shot predictions effectively sets upper bounds for anti-forgetting ability, reinforcing the theoretical foundation that zero-shot prediction stability can predict anti-forgetting capabilities.
This refined explanation and empirical data support the thesis that zero-shot stability is an effective metric for evaluating a model's ability to maintain learned knowledge over sequential tasks.

**Q2: More details on the motivation behind the loss term in Eq.(4), and what causes predictions for previous tasks to improve when learning a new task**

**A2:** As theoretically analyzed in the bounds of Proposition 1, both zero-shot stability and anti-forgetting ability are influenced by **three key factors**: the empirical error of continual tasks $\hat{\mathcal{E}}$, the discrepancy between task distributions $\rm{Div}$, and the complexity of the parameter space $\sqrt{\cdot}$. These factors should be minimized as much as possible to maximize the consistency between zero-shot stability and anti-forgetting ability.

In practice, this requires the implementation of carefully designed loss functions. As indicated in Eq.(4) of our manuscript:
  - The first term, $L_{CE}(P^t,\bar{P})$, aims to **minimize the empirical error** of continual tasks.
  - The second term, $L\_{ZS}(P^t, \widehat{{P}^t})$, focuses on **minimizing the task discrepancy**, which is both crucial and intriguing. Specifically, to make the discrepancy $\sum\_{i=1}^{t} {\rm{Div}}(\mathcal{T}\_\textit{i}, \mathcal{T}\_\textit{k})$ approach zero, we encourage the distribution of task $\mathcal{T}^{k}$ to converge towards the tasks $\{\mathcal{T}^{i}\}\_{i=1}^{t}$ in the semantic space. Thus, we aim to align its predictions $P^t$ from the current LoRA model with the predictions $\widehat{{P}^t}$ derived from the EMA-LoRA model on tasks $\{\mathcal{T}^{i}\}\_{i=1}^{t-1}$.
  - To explicitly **minimize the model complexity**, we have implemented our method within a parameter-efficient EMA-LoRA architecture, as opposed to utilizing full-parameter fine-tuning.

Improving predictions for earlier tasks when introducing a new task is challenging due to the **low discrepancy/high similarity** required between task distributions. We attempt to demonstrate this under ideal conditions, but such similarity is rare. Thus, our primary focus is on maintaining performance on previous tasks (anti-forgetting), a more achievable goal in continual learning.

**Q3: Relation to existing EMA-style objective, such as[1,2]**

**A3:** Our approach differentiates from traditional EMA-style methods[1,2] by incorporating EMA-LoRA **not only during inference but also throughout training**. These weights are integrated into objective optimization as part of the $L_{ZS}(P^t, \widehat{{P}^t})$ equation, functioning as a zero-shot stability regularizer. This integration **enhances performance** as demonstrated in our ablation studies (Table 3), where EMA-LoRA with $L_{ZS}$ outperforms traditional configurations.

Additionally, using EMA weights during training **reduces sensitivity to momentum parameters** (L312-319). This was tested across various settings, showing robust performance with varying $\alpha$ values:
|$\alpha$|0.50|0.55|0.60|0.65|0.70|0.75|0.80|0.85|0.90|
|-|-|-|-|-|-|-|-|-|-
|EMA-LoRA |18.70|20.18|20.10|19.93|19.64|19.03|18.12|17.30|13.73|4.74|
|ZAF|8.07|7.26|6.59|6.41|8.45|5.08|4.27|3.38|1.99|1.43

In summary, our EMA-style objective, highlighted as one of our contributions, is indeed implemented effectively, as shown in the comparison table below:
|Metrics|Existing EMA-style Works|Our Work|
|-|-|-|
|Training|$\textcolor{green}{\times}$|$\textcolor{red}{\checkmark}$|
|Inference|$\textcolor{red}{\checkmark}$|$\textcolor{red}{\checkmark}$|
|Performance|worse|better|
|Momentum insensitivity|$\textcolor{green}{\times}$|$\textcolor{red}{\checkmark}$|

[1]Momentum-based Weight Interpolation of Strong Zero-Shot Models for Continual Learning

[2]A unified continual learning framework with general parameter-efficient tuning

---

### Decision · Program_Chairs · 2024-09-25

**Decision:**

Accept (poster)

**Comment:**

The paper proposes a method to adapt pretrained vision language models to new tasks in a continual learning manner.
The reviewers appreciated the contribution of the proposed replay free regularizer and the combination of EMA  and LoRA.
The main concerns of reviewers have been cleared during rebuttal and all reviewers have recommended acceptence.